# Carbon and health implications of trade restrictions

Jintai Lin [1,14]*, Mingxi Du[1,14], Lulu Chen[1,14], Kuishuang Feng [2,3]*, Yu Liu[4,5]*, Randall V. Martin [6,7,8], Jingxu Wang[1], Ruijing Ni[1], Yu Zhao[9], Hao Kong[1], Hongjian Weng[1], Mengyao Liu[1], Aaron van Donkelaar[6,7], Qiuyu Liu[10] & Klaus Hubacek [11,12,13]

In a globalized economy, production of goods can be disrupted by trade disputes. Yet the resulting impacts on carbon dioxide emissions and ambient particulate matter ($PM_{2.5}$) related premature mortality are unclear. Here we show that in contrast to a free trade world, with the emission intensity in each sector unchanged, an extremely anti-trade scenario with current tariffs plus an additional 25% tariff on each traded product would reduce the global export volume by 32.5%, gross domestic product by 9.0%, carbon dioxide by 6.3%, and $PM_{2.5}$-related mortality by 4.1%. The respective impacts would be substantial for the United States, Western Europe and China. A freer trade scenario would increase global carbon dioxide emission and air pollution due to higher levels of production, especially in developing regions with relatively high emission intensities. Global collaborative actions to reduce emission intensities in developing regions could help achieve an economic-environmental win-win state through globalization.

[1] Laboratory for Climate and Ocean-Atmosphere Studies, Department of Atmospheric and Oceanic Sciences, School of Physics, Peking University, Beijing 100871, China. [2] Institute of Blue and Green Development, Shandong University, Weihai 264209, China. [3] Department of Geographical Sciences, University of Maryland, College Park, MD 20742, USA. [4] Institutes of Science and Development, Chinese Academy of Sciences, Beijing 100190, China. [5] School of Public Policy and Management, University of Chinese Academy of Sciences, Beijing 100049, China. [6] Department of Energy, Environmental & Chemical Engineering, Washington University in St. Louis, St. Louis, Missouri, United States. [7] Department of Physics and Atmospheric Science, Dalhousie University, Halifax, NS B3H 4R2, Canada. [8] Smithsonian Astrophysical Observatory, Harvard-Smithsonian Center for Astrophysics, Cambridge, MA 02138, USA. [9] School of the Environment, Nanjing University, 163 Xianlin Ave, Nanjing 210046, China. [10] Department of Biological Sciences, University of Quebec at Montreal, Montreal H3C 3P8, Canada. [11] Energy and Sustainability Research Institute Groningen (ESRIG), University of Groningen, Nijenborg 6, 9747 AG Groningen, The Netherlands. [12] International Institute for Applied Systems Analysis, Schlossplatz 1, A-2361 Laxenburg, Austria. [13] Department of Environmental Studies, Masaryk University, Jostova 10, 602 00 Brno, Czech Republic. [14] These authors contributed equally: Jintai Lin, Mingxi Du, Lulu Chen. *email: linjt@pku.edu.cn; kfeng@umd.edu; liuyu@casipm.ac.cn

Economies worldwide are tightly connected through trade. Goods and services are consumed and produced in different parts of the world often with different resource availability, economic and energy structures, environmental regulations, and meteorological and chemical conditions[1]. Recent work based on empirical trade data has shown that, with given amounts of global total burdens, large quantities of carbon dioxide ($CO_2$), particulate matter ($PM_{2.5}$) pollution, and related premature deaths are embedded in traded products[2–6]; in other words, these environmental burdens are associated with production for export. However, whether trade improves or worsens environmental quality depends on the level of progress in the economy, regulation, and technological development[7–9], which varies along global supply chains. In part as a reaction to perceived disadvantages caused by trade of certain goods and services, the past years have seen a number of attempts to restrict trade activities[10–13]. Yet, the grand question of whether trade liberalization or restriction leads to a better global environment remains unclear.

Here, we assess the potential impacts of trade restrictions on $CO_2$ emissions and $PM_{2.5}$-related premature mortality at both global and regional scales. The assessment is done based on both economic and emission data in 2014, the latest year for which all necessary data are available. Based on five trade scenarios differentiated by the extent of trade restrictions, we take an interdisciplinary approach to integrating the latest standard Global Trade Analysis Project (GTAP, version 10 data base for 2014), a computable general equilibrium (CGE) model[14–16] for global trade and economic analysis, a customized emission inventory derived from the Community Emission Database System (CEDS)[17] and Xia et al.[18], the GEOS-Chem atmospheric chemical transport model[19], a satellite-based dataset for near-surface $PM_{2.5}$ mass concentrations[20], and the Global Exposure Mortality Model (GEMM)[21] for pollution exposure (see Methods for details). Emissions and premature deaths analyzed here are only those which are linked to changes in economic output of 20 industrial sectors and 13 aggregated regions associated with each trade scenario. Scenario-dependent $PM_{2.5}$ considered here include secondary inorganic aerosols (SIOA, including sulfate, nitrate, and ammonium), black carbon (BC), and POA. We find substantial impacts of trade restrictions on the global magnitude and regional distribution of emissions and health burdens.

## Results and Discussion

**Global free trade (GFT) scenario**. This scenario assumes zero border tax for all traded products. It leads to the highest global export volume, GDP, $CO_2$ emissions, and premature deaths (Fig. 1a–d). As simulated by the CGE model[16], the global export volume reaches 22.1 trillion and GDP reaches $79.3 trillion in US Dollar in 2014. Western Europe, the US and China contribute, respectively 33.7%, 8.7%, and 15.0% of the global export volume and 24.4%, 22.1%, and 14.4% of global GDP (Fig. 1a, b). Global $CO_2$ emission amounts to 25.6 Petagram (Pg), of which 57.8% are contributed by China (29.6%), the US (16.4%), and Western Europe (11.8%) (Fig. 1c). Note that global emissions do not include scenario-independent sources, which together are about 9.9 Pg (see Methods for details).

The regional distribution of pollutant emissions reveals a different picture (Supplementary Fig. 1). China, South Asia, and Middle East and North Africa are the top three emitters, and they together contribute 47.5–60.2% of the global emissions of sulfur dioxide, nitrogen oxides, ammonia, carbon monoxide, BC, and POA. A major driver of the large amount of emissions in these regions is their high-emission intensities (i.e., emissions per monetary output). Supplementary Fig. 2 shows that emission intensities in these regions are about 7–23 times of those in Western Europe and the USA (pollutant dependent). In general, emission intensities in developing regions are much larger than those in developed regions, and regions with higher per capita GDP tend to have lower emission intensities (Supplementary Fig. 2).

High emissions in many developing regions contribute to their heavy $PM_{2.5}$ pollution (Supplementary Fig. 3), in addition to the influences of meteorological and chemical conditions, as simulated by GEOS-Chem[19]. China and South Asia experience the highest anthropogenic, population-weighted $PM_{2.5}$ concentrations (22.6–23.9 µg/m$^3$, for scenario-dependent SIOA, BC, and POA together). For China and South Asia, their atmospheric conditions are also favorable for local pollution accumulation, i.e., the chemical efficiency of their emissions to form and accumulate $PM_{2.5}$ locally are high (Supplementary Fig. 4). By comparison, favorable atmospheric conditions for South-East Asia and Pacific allow their pollution to be more quickly deposited to the ground or transported out of their territories, contributing to their relatively low $PM_{2.5}$ concentrations. Nonetheless, atmospheric transport allows regionally emitted/formed pollution to be transferred to vast downwind areas (Supplementary Fig. 4).

In Scenario GFT, anthropogenic $PM_{2.5}$ pollution (SIOA, BC, and POA together) leads to a large number of premature deaths worldwide (Fig. 1d, Supplementary Fig. 3). Based on the GEMM NCD+LRI pollution-health response model[21], the number of deaths reaches 2.94 million [95% CI: 1.72–4.14] globally, 1.02 million [95% CI: 0.61–1.43] in China, 0.89 million [95% CI: 0.49–1.29] in South Asia, 0.29 million [95% CI: 0.17–0.41] in Western Europe, and 0.11 million [95% CI: 0.07–0.15] in the USA. The high values in China and South Asia are also due to their large baseline mortality (8.5 and 6.8 million, respectively).

**Actual trade restriction scenario (ATR)**. This scenario represents the actual tariff situation in 2014. Compared to GFT, it has a global average border tax of about 5% (see Supplementary Data 1 for regional details). This leads to reductions by about 5.4% in the global export volume ($1.19 trillion), 1.3% in GDP ($1.05 trillion), 1.2% in emissions (317.5 Tg for $CO_2$), and 1.1% in $PM_{2.5}$-related premature mortality (32.7 thousand) (Fig. 1e–h). The most affected region is Japan and Korea, whose export volume is reduced by 9.7%, GDP by 3.1%, $CO_2$ emission by 3.7%, and premature mortality by 2.1%. This is because of the region's large dependence on trade. The impacts for China and South Asia are larger than the global average: by 10.7–17.4% for the export volume, 2.1–2.6% for GDP, 1.3–2.0% for $CO_2$, and 1.0–1.3% for mortality. By comparison, the impacts on the US and Western Europe are smaller (by 1.2–4.0% for export volume, 0.7–0.9% for GDP, 0.75–0.84% for $CO_2$, and 0.7–0.9% for mortality).

**Sino-US trade war scenarios (TW1 and TW2)**. These two scenarios represent the increasing levels of bilateral trade wars between the US and China. Scenario TW1 represents the stage of the Sino-US trade war by the end of 2018 where, on top of ATR, the United States imposes additional border taxes for $250 billion worth of products imported from China, while China imposes extra tariffs for $110 billion worth of imported US products. Scenario TW2 represents a hypothetical full-blown stage of the Sino-US trade war where, on top of ATR, the two countries impose an additional 25% tariff on any product imported from the other country. As expected, the Sino-US trade war scenarios (TW1 and TW2) have lower amounts of carbon emissions and premature deaths than the GFT scenario for all regions (Fig. 1i–p). Compared to Scenario GFT, the global export volume is reduced by 5.8% in TW1 and 6.1% in TW2, GDP by 1.46% and

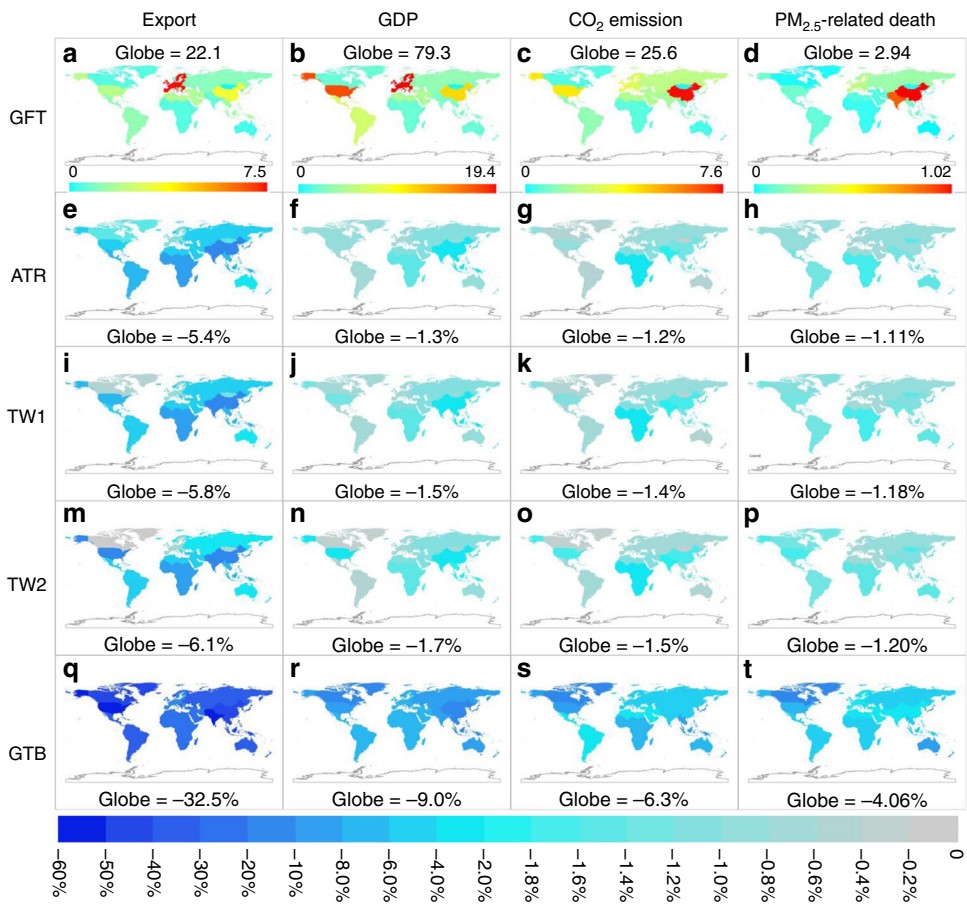

**Fig. 1** Economic volume and environmental quality in each scenario. **a–d** Regional export volume (trillion $), GDP (trillion $), $CO_2$ emission (Pg), and $PM_{2.5}$ related premature death (million) in Scenario GFT. **e–t** Relative changes from global free trade (GFT) scenario to each alternative scenario. Results here only include scenario-dependent sources. Note that the color scales are linear in (**a–d**) and nonlinear in (**e–t**)

1.67%, $CO_2$ emission by 1.39% and 1.50%, and premature mortality by 1.18% and 1.20%, respectively.

Compared to ATR (Supplementary Fig. 5), the two trade war scenarios reduce GDP, $CO_2$ emission, and mortality in both the USA (by 0.33–0.39% for TW1, 0.76–1.26% for TW2) and China (by 0.18–0.64% for TW1, 0.24–1.22% for TW2), but with increases in other regions. The most significant GDP increases are in the rest of North America, Japan and Korea, South-East Asia and rest of East Asia, because of their strong competitiveness in producing products targeted in the Sino-US trade war. Globally, the GDP, $CO_2$ emission, and mortality are also reduced by 0.07–0.15% for TW1 and 0.09–0.36% for TW2. The changes in global and regional GDP are consistent with other studies (Supplementary Table 1).

**Global trade barrier (GTB) scenario.** The GTB scenario represents a world in which every region has significant trade barriers. Compared to the GFT scenario, it leads to a substantial reduction by 32.5% in global export volume and 9.0% in GDP. The regional impacts are even more significant (Fig. 1q, r). Of the 13 aggregated regions, the US has the greatest reduction in export volume (57.2%), and South-East Asia has the greatest reduction in GDP (16.0%). The export volume of Western Europe, the US, South Asia, and China would decrease by 11.7–57.2% and GDP by 6.7–10.9%.

When moving from the GFT scenario to the most restrictive GTB scenario, global emissions are reduced by 6.3% for $CO_2$ and 4.7–6.3% for the six air pollutants. The regional impacts are broadly consistent with the impacts on GDP (Fig. 1s, t, Supplementary

Fig. 6), although there are substantial differences due to regional and sectoral disparities in emission intensity. For Western Europe, the US, and China, $CO_2$ emissions are reduced by 4.9%, 8.2%, and 5.4%, respectively, and emissions of air pollutants are reduced by up to 3.3%, 8.4–10.0%, and 2.6–4.7%, respectively (Supplementary Fig. 6).

From GFT to GTB, $PM_{2.5}$-related premature mortality decreases by 119 thousand (or 4.1%). This value is larger than the number of total premature deaths in the US, rest of North America, and Oceania due to exposure to ambient $PM_{2.5}$ in GFT. South Asia (35.5 thousand) and China (33.3 thousand) have the large absolute reductions in premature mortality. The rest of North America (10.3%), Japan, and Korea (9.1%), Oceania (9.0%), and the USA (7.7%) have the greatest relative reductions, that is, more than twice the global average reduction.

**Synergy of all scenarios.** Figure 2 presents the relative changes in regional $CO_2$ emission and mortality as a function of GDP change across the individual scenarios relative to the GFT. In general, as the trade restrictions tighten from GFT to the actual trade in 2014 (ATR), to the Sino-US trade war scenarios (TW1 and TW2), and finally to the GTB scenario, regional GDP, $CO_2$ emission, and mortality also decrease. However, there exist substantial regional differences in this relationship, as apparent from the scatter plot in Fig. 2. Overall, about 52–64% of global $CO_2$ emission reduction and 78–83% of global mortality reduction from GFT to ATR, TW1, TW2, and GTB occur in developing regions (China, rest of East Asia, Economies in Transition, Latin America and Caribbean, Middle East and North Africa, South

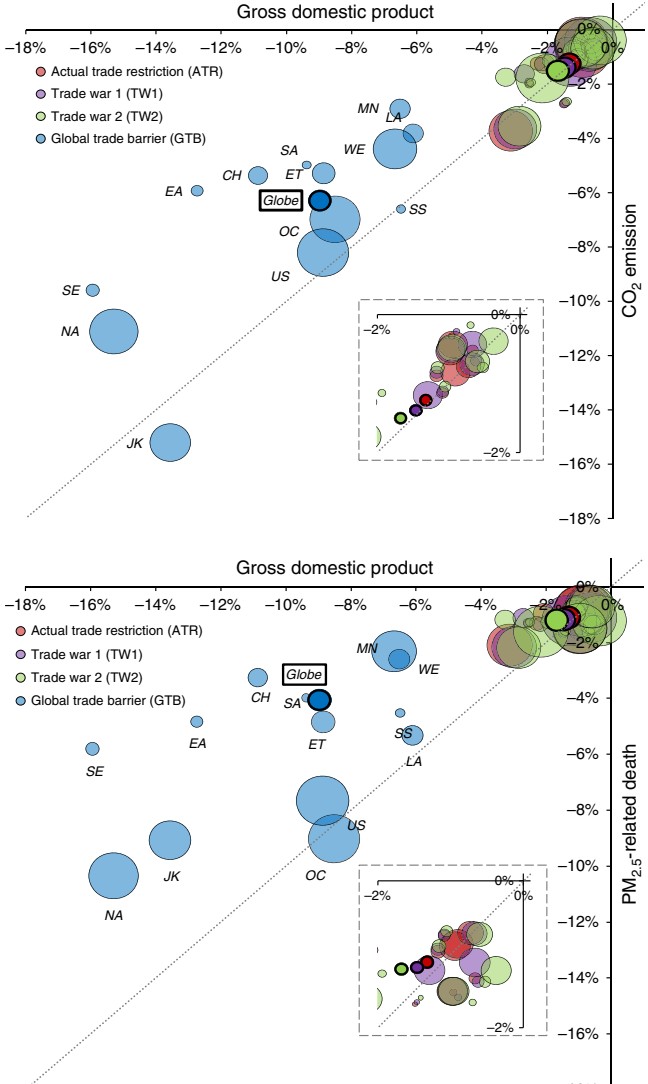

**Fig. 2** Contrasting changes in economic performance and environmental quality. The Figures show the relative changes in regional $CO_2$ emission and $PM_{2.5}$ related premature mortality from global free trade scenario (GFT) to each alternative scenario, as a function of respective changes in regional GDP. The color of the dots denotes individual trade scenarios, and the size of the dots denotes the magnitude of regions' per capita GDP. The dots with darker colors and thicker borders denote the global values. The inserted Figures in dotted boxes are zoom-in ones to indicate ART, TW1, and TW2. Results here only include scenario-dependent sources. Regions include China (CH), rest of East Asia (EA), Economies in Transition (ET), Japan and Korea (JK), Latin America and Caribbean (LA), Middle East and North Africa (MN), rest of North America (NA), Oceania (OC), South Asia (SA), South-East Asia and Pacific (SE), Sub-Saharan Africa (SS), the United States (US), and Western Europe (WE)

Asia, South-East Asia and Pacific, and Sub-Saharan Africa), with the rest in developed regions (Fig. 3).

At the global level and for most regions, the relative reductions from GFT to GTB in terms of $CO_2$ emissions and mortalities are less significant than the reduction in GDP. This means that restricting trade is not an effective approach for reducing emissions. This result also indicates enhanced (sectorally averaged) emission intensities of $CO_2$ and pollutants in an antitrade world represented by Scenario GTB, compared to GFT. This is because individual economic sectors have different

emission intensities[22,23] and different responses to economic shocks from trade restrictions. Sectors with high emission intensities such as Electricity and Road Transport are often not directly affected by trade restrictions, since they do not produce goods for trade. By comparison, sectors with low emission intensities, such as Wearing Apparel and Textiles are often directly affected by trade restrictions. As shown in Supplementary Fig. 7, the relative reduction in economic output from GFT to GTB is smaller in more emission-intensive sectors, resulting in increased relative contributions of emission-intensive sectors to global output.

For a given amount of relative reduction in GDP from GFT to GTB, developed regions tend to have greater relative reductions in mortality than developing regions do (Fig. 2). This is because in the more protected environments of developed regions, a given amount of pollution concentration change is more conducive to change in mortality risk[21].

When moving from a GFT to a restricted trade scenario (GTB), the global export volume would decrease by 32.5%. This means that there are still significant amounts of interregional trade activities in GTB. We do not simulate a situation in which all trade activities are banned, which would be highly unlikely to occur. Nonetheless, assuming a linear relationship between the change in global export volume and changes in $CO_2$ emission and mortality, we estimate that fully banning trade would lead to reductions in global GDP, $CO_2$ emission, and mortality by 27.0%, 18.9%, and 12.3%, respectively, compared to a world with a larger extent of free trade.

**Decoupling environmental impacts from trade liberalization**. The above results imply that with fixed sectoral emission intensities, trade liberalization scenarios may lead to an improved global economy but at a larger environmental cost. These environmental side effects come with a higher amount of production, the influence of which is partially compensated by a reduction in sectorally averaged, global mean emission intensity due to changes in trade pattern and economic structure (Supplementary Fig. 7). Changes in $CO_2$, pollutant emissions, and mortality are dominated by those in developing regions with higher emission intensities (Fig. 3). Thus, reducing emission intensities in developing regions is key to alleviating adverse environmental consequences of trade liberalization.

High emission intensities in developing regions are caused by multiple factors. Developing economies tend to rely on fossil fuel, especially coal, much more than developed economies, because of more limited access to cleaner or renewable energy sources which are usually more expensive and/or technologically challenging[24,25]. Meanwhile, developing economies are shifting toward producing emission-intensive goods whereas developed economies are shifting away[26,27]. Developing regions also have looser environment regulations and enforcement and lower energy and product use efficiencies, due in part to lack of advanced technology and know-how[28,29].

Global collaborative efforts can be made to reduce emission intensities in developing regions. This is of global value given the global climate impact of $CO_2$ and the transboundary atmospheric transport of air pollutants[3,6]. The Paris Agreement has already included technological and financial support to developing regions[30,31]. Implementing and enhancing these aids would be valuable to alleviate the reliance of developing regions on fossil fuels, improve their energy and production efficiencies, and enhance their emission control capabilities. These actions might be accompanied by negotiations on moving toward more consistent environmental standards and policies (e.g., carbon pricing) across the globe[32].

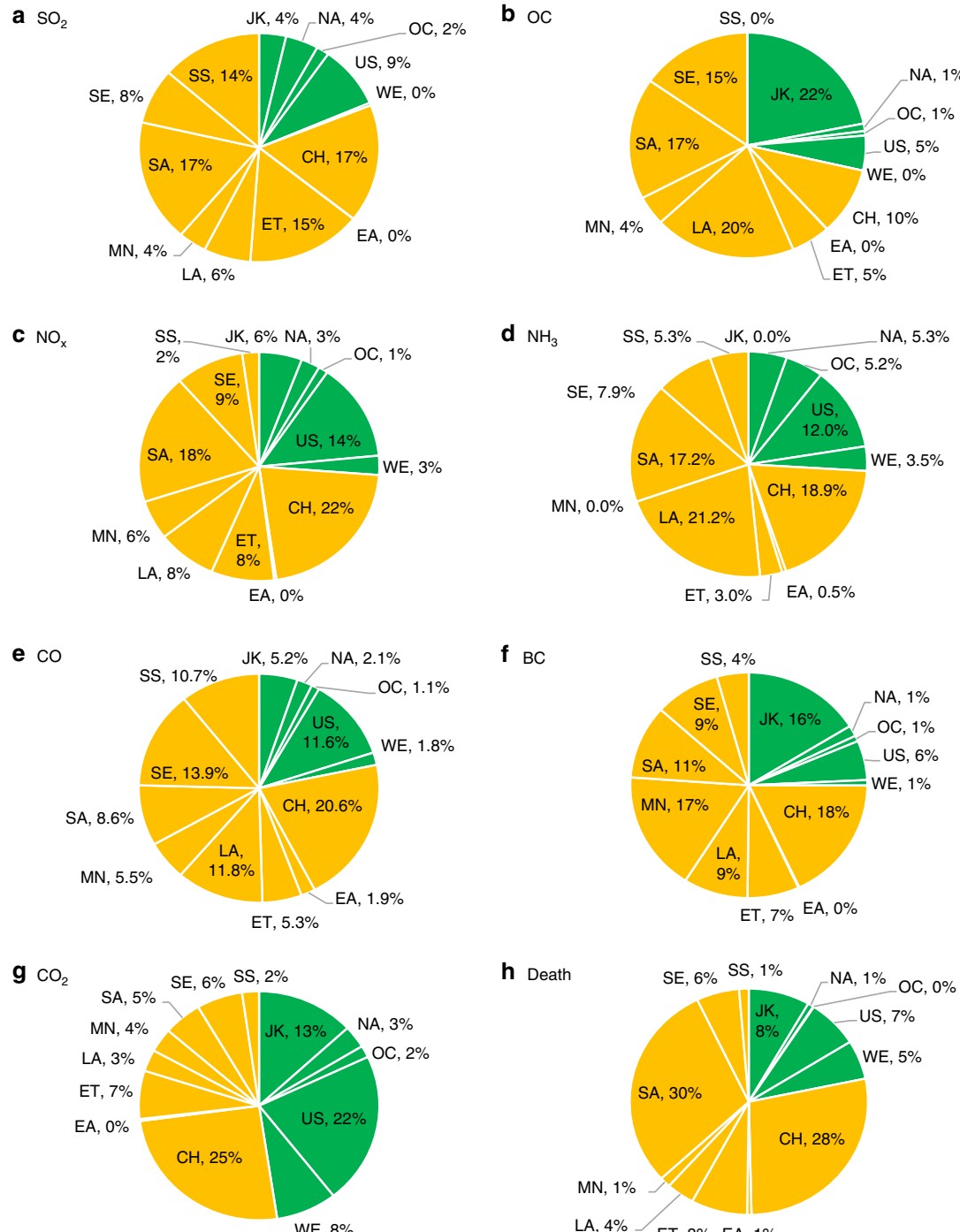

**Fig. 3** Regional contributions to global changes in (**a–f**) pollutant emissions and (**g**) $CO_2$ and (**h**) $PM_{2.5}$ related mortality from global free trade scenario (GFT) to global trade barrier (GTB) scenario. The green color represents developed regions, and the orange color represents developing regions. Results here only include scenario-dependent sources. Regions include China (CH), rest of East Asia (EA), Economies in Transition (ET), Japan and Korea (JK), Latin America and Caribbean (LA), Middle East and North Africa (MN), rest of North America (NA), Oceania (OC), South Asia (SA), South-East Asia and Pacific (SE), Sub-Saharan Africa (SS), the United States (US), and Western Europe (WE)

In order to estimate how trade liberalization can be accompanied by an improved global environment, we contrast GFT against an additional scenario (GFTT) which assumes global free trade plus sufficient global technological/financial support and more globally consistent environmental policies to further reduce emission intensities in developing regions. Enhancing environmental regulations in developing regions means an economic burden (at least in the beginning) to industries that may affect their competitiveness and subsequently alter

interregional trade, which is not fully accounted for in GFTT. Nonetheless, enhancing the financial and technological support may reduce the initial shock to developing economies. Under the GFTT scenario, the emission intensity of a sector in any region that is higher than the global sectoral average is reduced to the average value. As a result, global $CO_2$ emission would be reduced by 24.2%, pollutant emissions by 27.3–53.6%, and $PM_{2.5}$-related premature deaths by 36.0%. The respective regional reductions are substantial (Fig. 4). For example, $CO_2$ emission would be

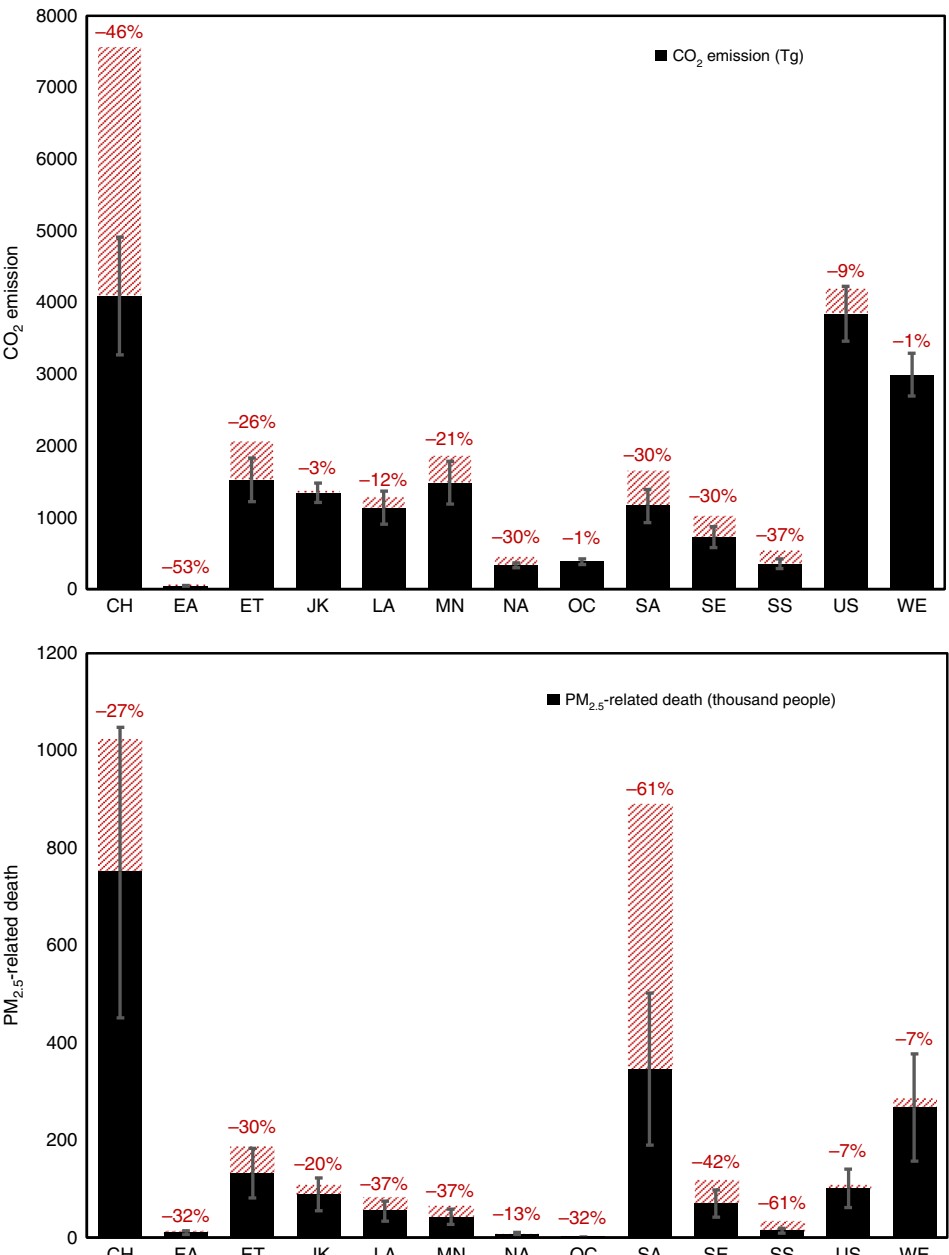

**Fig. 4** Regional $CO_2$ emission, $PM_{2.5}$-related premature mortality under global free trade plus technological/financial support scenario (GFTT). Error bars denote uncertainty ranges (95% CI). Red dashed bars denote the amounts of respective reductions from global free trade scenario (GFT). Results here only include scenario-dependent sources. Regions, include China (CH), rest of East Asia (EA), Economies in Transition (ET), Japan and Korea (JK), Latin America and Caribbean (LA), Middle East and North Africa (MN), rest of North America (NA), Oceania (OC), South Asia (SA), South-East Asia and Pacific (SE), Sub-Saharan Africa (SS), the United States (US), and Western Europe (WE)

reduced by 46.0% in China and 30.4% in South Asia, and $PM_{2.5}$-related mortality would be reduced by 61.1% in South Asia and 60.8% in sub-Saharan Africa.

Several sources of uncertainty and limitation exist in our study, as detailed in Methods. The standard GTAP model is an equilibrium model that does not simulate the temporal evolution of the economies. Emission data are subject to large errors especially for air pollutants. We do not account for the change in emission intensity of a given sector from one trade scenario to another, although the overall sectorally averaged emission intensity (i.e., total emission divided by total output from all sectors) is allowed to change because of the change in sectoral output structure. GEOS-Chem simulations are subject to errors in emissions and model representations of atmospheric

chemical and physical processes. In particular, secondary organic aerosols (SOA)[33–35] are not simulated here, considering the relative small contribution of anthropogenic SOA to the total $PM_{2.5}$[36–39]. Using chemical efficiencies to calculate pollution levels for each trade scenario further introduces a minor source of error. For each scenario, a major source of error arises from the pollution-health exposure models used here. Quantitative estimates of these errors are given in Methods. The overall error results are expressed as 95% CI in the main text. Although errors in emissions and pollution-health models are dominant, they are derived from causes that do not depend on trade scenarios, and are thus not relevant when discussing the relative change in premature mortality from one trade scenario to another.

Over the past few decades, trade has been associated with movement of pollution-prone economic production from developed to developing regions[26]. This movement is continuing as production is being relocated from wealthier to poorer, developing regions[40]. Given the substantial gap in emission intensity between developed and developing regions, this movement might have meant a less environmentally friendly global economy, causing an unnecessary dilemma between trade-associated economic development and environmental sustainability[41]. Eliminating this dilemma will require substantial reductions in emission intensities in developing regions, crucially through local effort, global collaboration, and other collective action against environmental degradation. To this end, our study offers insight for policymakers who might consider to better address in trade negotiations the potential environmental implications of trade to ensure sustainable growth on regional and global scales.

## Methods

**An interdisciplinary framework.** Our interdisciplinary approach to calculating the carbon and pollution health impacts of trade restrictions consists of a few steps. Supplementary Fig. 8 shows the overall framework.

First, we design five scenarios ranked by the extent of trade restrictions between 31 regions across 20 industrial sectors worldwide. Second, we use the GTAP computable general equilibrium (CGE) model[14,16] to simulate sector-specific interregional trade flows as well as other economic indicators and $CO_2$ emissions.

Third, we calculate anthropogenic emissions of air pollutants specific to each trade scenario, by combining GTAP-modeled scenario-specific economic output and a prescribed, scenario-invariant dataset of emission intensities (i.e., emissions per monetary unit of economic output). The prescribed emission intensity data are specific to each pollutant in each of the 20 sectors in 31 regions, and are calculated based on economic output data in Scenario ATR (that represents the actual global economy in 2014) and a customized anthropogenic emission inventory. The inventory is from the CEDS[17], with some improvements for China[18,42–44].

Fourth, we use simulations of the GEOS-Chem model[19] to derive near-surface $PM_{2.5}$ mass concentrations for individual trade scenarios. The 31 emission source regions above are further aggregated into 13 regions (Supplementary Fig. 9) to reduce computational costs.

Finally, we apply the $PM_{2.5}$ concentrations derived above to the GEMM[21] to evaluate the health impacts in each trade scenario. Mortality results based on the integrated exposure-response model (IER)[45] are also presented in Supplementary Data 2 for comparison. Results for the 13 aggregated regions are discussed in the main text.

**Trade scenarios.** Scenario GFT represents a world with no trade restrictions. In this scenario, the tariffs on all commodities in all regions are set to zero.

Scenario ATR represents the Actual Trade Restriction in 2014, according to the tariff and other economic data in the GTAP v10 database[15,46]. Supplementary Table 1 shows the tariff setting.

Scenario TW1 represents the stage of the Sino-US trade war as of 2018. At this stage, the US imposes a 25% additional tariff on 728 specific products imported from China that are worth $50 billion in total[47], and a 10% additional tariff on 5745 items imported from China that are worth $250 billion together[48]. Meanwhile, China imposes a 25% additional tariff on 659 specific products imported from the USA that together are worth $50 billion[10,49], and a 10% additional tariff on 2493 + 1078 items and a 5% additional tariff on 974 + 662 items imported from the USA that together are worth $60 billion[11].

Thus, Scenario TW1 assumes that on top of Scenario ATR, the USA imposes additional tariffs on about $274 billion worth of products imported from China, and China imposes additional tariffs on about $116 billion worth of products from the USA. Supplementary Table 2 shows the tariff setting.

Eq. (1) shows how the sector-specific tax rates are converted from product-based rates. In reality, all products are classified based on the eight-digit subheadings of the Harmonized Tariff Schedule of the USAs and China. In our study, products are classified based on the 6-digit subheadings from UN Comtrade Database[50] and then mapped to the 20 industrial sectors. Because of this product-to-sector conversion, the monetary volumes of imported products with imposed tariffs in this scenario ($274 billion and $116 billion) are slightly different from the actual volumes ($250 billion and $110 billion). Another likely cause of such differences is that we use the actual trade data in 2014 rather than those in 2018, with slightly different trade and tariff information. The imposed tariff on each industrial sector in GTAP is calculated by

$$T = V'/V \times T',$$ (1)

Here, $T'$ and $V'$ denote the imposed tax rate and associated trade volume for each product belonging to a particular sector studied here. $T$ and $V$ denote the imposed tariff and associated trade volume in each of the 20 sectors studied here. Data of $T'$

(25%, 10%, or 5%) are taken from reports by the USA Trade Representative and the Ministry of Finance of the People's Republic of China[10,11,47–49]. Data of $V'$ and $V$ are from the UN Comtrade Database[50].

Scenario TW2 represents the potential full-blown stage of the Sino-US trade war. This scenario, which is built upon scenario ATR, assumes that the Sino-US trade war intensifies to the extent that both the US and China impose an additional 25% tariff on all goods imported from each other. Supplementary Table 2 shows the tariff setting.

Scenario GTB represents a world in which every region moves strongly against trade such that on top of Scenario ATR, each region imposes an additional 25% tariff on all products imported from all other regions. Supplementary Table 2 shows the tariff setting. This scenario is highly unlikely to happen in the near future. Nonetheless, the on-going anti-globalization movement in many countries[51,52] suggests that intensive trade wars might also occur between countries other than the USA and China, providing some rationale for this extreme scenario.

**The GTAP model.** The GTAP CGE model is a multiregional, multi-sector economic equilibrium model. With a long history of systematic improvements, GTAP provides an effective tool for a variety of studies related to trade, the environment, population, energy, and climate change[53–60]. The model is a comparative static analysis model, assuming that the market is completely competitive and the returns to scale of production remain unchanged[14]. Taking these theoretical assumptions, producers are assumed to maximize profits, while consumers maximize their utility. The total supply and total demand are in equilibrium, and they together determine the values of endogenous variables, such as price, wages, and land rents. All economies (countries and regions) connect with each other through commodity trade.

Each production activity is a combination of intermediate goods and factors to produce output. Similar to many CGE models, the production structure inside GTAP is based on a sequence of nested constant elasticity of substitution (CES) functions that aims to reproduce the substitution possibilities across the full set of inputs. The top-level nest is composed of two aggregate composite bundles, i.e., intermediate demand and value added. The second level nest decomposes each of the two aggregate composite bundles into their components, such that one is demand for individual intermediate goods and the other is demand for primary factors. The final nest accepts the Armington assumption to allow an incomplete substitution between domestically produced goods and imported goods.

For private households, the particular functional form chosen here to represent preferences is based on the constant differences of elasticities implicit additive expenditure function by Hanoch[61]. The Cobb–Douglas function is adopted for depiction of government consumption. The sub-utility function for investment expenditure, i.e., gross investment, is based on a Leontief utility function. The aggregate volume of investment comes from the nominal investment equals saving identity, where saving is the sum of domestic saving and net capital inflows from foreign economies. Investment expenditures on the composite goods are subsequently decomposed into demand for domestic and imported goods using a CES sub-utility preference function.

In addition, GTAP includes five types of primary factors including land, capital, skilled labor, unskilled labor, and natural resources, and three representative agents including private households, governments, and companies. Within each country or region, the GTAP model allows capital and labor to move between production sectors, and partially allows land to move between crop producing sectors based on the CET assumptions.

The GTAP CGE model used here is built based on the latest version (v10) of the GTAP database[15], which is constructed from the input–output tables of 141 countries and regions across the world with a base year of 2014. The GTAP database contains 57 sectors and 5 primary production factors. For this study, the 141 countries and regions have been aggregated to 31 regions, which specify major producers, consumers, and importers/exporters (see the mapping in Supplementary Data 3). The 57 production sectors are aggregated to a total of 20 sectors (see the mapping in Supplementary Data 3). The five types of original primary factors have been aggregated to three categories (land, capital, and labor).

**Anthropogenic emissions of $CO_2$.** For each scenario, anthropogenic emissions of $CO_2$ are computed from the GTAP model. These emissions are calculated based on sector-specific emission factors (embedded in the GTAP database and unchanged across the trade scenarios) and scenario- and sector-specific energy consumption data computed from the GTAP model.

Anthropogenic emissions considered here are due to fuel combustion associated with economic production (i.e., those which directly produce GDP), except international shipping and aircraft emissions. These emissions vary from one trade scenario to another. Although different trade scenarios may affect emissions from international shipping and aircraft, these emissions are not accounted for here, due to lack of robust methods to allocate these emissions to specific regions. Emissions from residential activities (such as heating and cooking at home) and private transport are not included here. The process-related emissions (such as cement production) are not included here. Together, the sources not analyzed here contribute 9.9 Pg of global $CO_2$ emission[15,62].

**Anthropogenic emissions of air pollutants**. The GTAP model does not provide emissions of air pollutants. Thus, we calculate anthropogenic air pollutant emissions for each trade scenario based on prescribed sector-, region-, and pollutant-specific emission intensity data (that remain unchanged across the trade scenarios) and scenario-specific economic output from GTAP.

We derive the sector-, region-, and species-specific emission intensities by combining a customized emission inventory (CEDS+Xia, see below) in 2014 and economic output data in the GTAP database in 2014.

We use the monthly gridded (0.5° longitude × 0.5° latitude) CEDS inventory[17] for global anthropogenic emissions of gaseous ($SO_2$, $NO_x$, $NH_3$, NMVOC, and CO) and primary aerosol (BC and OC) pollutants worldwide in 2014, with 54 sectors in 152 regions. The CEDS inventory has a globally consistent and reproducible methodology applied to all pollutants and includes updated emission factors[17]. It provides very detailed sectoral emission information, which is essential for this study. The inventory is being used by the Coupled Model Intercomparison Project Phase 6 (CMIP6, a main model support for the Inter-governmental Panel on Climate Change Sixth Assessment Report) and many other studies[63–67].

The CEDS inventory uses regional emission data over the US (1990–2014), Canada (1990–2013), Europe (1980–2013), China (2008, 2010, and 2012), and other regions to revise its initial global methodology. Over the past decade, the amount of emissions in China varied significantly from 1 year to another due to implementation of stringent emission control measures as well as the changing economy and fossil fuel consumption. Thus, we replace the seasonal and spatial patterns of Chinese $SO_2$, $NO_x$, CO, BC, and POA emissions in CEDS by those in the MEIC inventory in 2014[68]. We further scale Chinese annual $SO_2$, $NO_x$, CO, BC, and POA emissions in CEDS to match those developed by Xia et al.[18,42–44], which account for the pollution control measures more comprehensively and may better represent the actual emissions in China in 2014. Since the CEDS and Xia et al. inventories contain different sectors, we conduct a sector mapping procedure (Supplementary Data 4). Hereafter, we refer to this hybrid inventory as CEDS+Xia.

The CEDS+Xia inventory contains 54 sectors in 152 countries/regions. Of these sectors, 8 belong to energy production, 23 belong to industry, 8 belong to transportation, 4 belong to residential, 5 belong to agriculture, 4 belong to waste treatment, and the remaining 2 represent other unspecified processes that are associated with very few emissions. The inventory is a global monthly gridded dataset at a 0.5° longitude × 0.5° latitude resolution, beneficial for subsequent GEOS-Chem simulations.

The CEDS+Xia inventory contains activities that produce significant amounts of emissions but do not directly produce economic output, i.e., these activities do not produce GDP directly and are not included in the GTAP model. In linking the emission inventory and GTAP, we exclude these activities to better quantify the sector-specific emission intensities. Nonetheless, these emissions are included in GEOS-Chem simulations to derive the total (anthropogenic + natural) $PM_{2.5}$.

First, emissions from the four sectors related to residential activities in CEDS +Xia are excluded, because these activities do not produce economic output accounted for in GTAP. This procedure was done also in previous studies[2,3,6].

The transportation sector in the CEDS+Xia inventory does not separate emissions associated with commercial vehicles from emissions associated with private vehicles. Private transport does not produce economic output accountable in GTAP, as opposed to commercial transport. Thus, we exclude emissions from private vehicles from the trade scenario analysis. This procedure improves upon previous studies that did not differentiate private and commercial transport[2,3,6].

To differentiate emissions associated with private and commercial transport, we use the vehicle emission data from the greenhouse gas and air pollution interactions and synergies (GAINS) model. The GAINS model provides transportation related emissions from four vehicle types, including passenger cars, light duty vehicles, heavy duty vehicles and buses, mopeds, and motorcycles (Supplementary Table 3).

We derive the contribution of private transport to the total transportation-related emissions as follows:

$$FC_{r,p} = \frac{\sum E_{r,s}^{pr}}{\sum E_{r,p}^{pr} + \sum E_{r,p}^{co}}. \qquad (2)$$

Here, $FC_{r,p}$ represents the fractional contribution of private vehicle driving for a given pollutant species p in a given region r. $E_{r,p}^{pr}$ and $E_{r,p}^{co}$ represent emissions of species p in region r from private and commercial vehicle driving, respectively, in the GAINS model.

The GAINS model only covers 74 regions. For a region with no GAINS data, $FC_{r,p}$ from its neighbor regions are employed. Supplementary Data 5 shows the fractional contribution of private vehicle driving $FC_{r,p}$ in each region.

Based on the CEDS+Xia inventory, we derive a prescribed dataset for emission intensity that varies across the sectors, regions and pollutant species, by dividing the CEDS+Xia emissions by the economic output data in the GTAP database for 2014. This emission intensity dataset is used and remains unchanged in all trade scenarios.

We convert the CEDS+Xia emissions for 152 regions and 54 sectors to 141 regions and 57 sectors according to the original GTAP setup, and then to 31 regions and 20 sectors to match those in our trade scenario analyses. The mapping details are shown in Supplementary Data 6.

Subsequently, we calculate emission intensity for each pollutant in each of the 20 sectors and 31 regions

$$F_{s,r,p}^b = E_{s,r,p}^b / X_{s,r}^b. \qquad (3)$$

Here, the subscripts s, r, and p denote the sector, region, and pollutant species, respectively. The superscript b denotes the base year (2014) that has CEDS+Xia emissions ($E_{s,r,p}^b$) and economic output in the GTAP database for 2014 ($X_{s,r}^b$).

For each trade scenario, trade scenario-dependent anthropogenic emissions ($E_{s,r,p}^{c,t}$) are derived from the prescribed, scenario-invariant emission intensity ($F_{s,r,p}^b$) and scenario-specific economic outputs ($X_{s,r}^c$)

$$E_{s,r,p}^{c,t} = X_{s,r}^c \times F_{s,r,p}^b. \qquad (4)$$

$$E_{r,p}^{c,t} = \sum_s E_{s,r,p}^{c,t}. \qquad (5)$$

Here, the subscripts s, r, and p denote the sector, region, and pollutant, respectively. The superscript c denotes the trade scenario, and t indicates that the emission is scenario-dependent. $E_{r,p}^c$ denotes the emission summed over all of the 20 sectors. $E_{r,p}^{c,t}$ does not include emissions from residential activities and private vehicles.

Emissions from international shipping and aircraft are taken from other sources. Although different trade scenarios may affect these emissions, they are kept constant here, due to lack of robust methods to allocate these emissions to specific regions. As such, emission and pollution changes from one trade scenario to another discussed in this study do not include the changes in international shipping and aircraft. These emissions are not discussed in the main text. Nonetheless, these emissions are used in GEOS-Chem simulations to derive the total (anthropogenic + natural) $PM_{2.5}$ discussed in next section.

**GEOS-Chem simulations**. Through a series of simulations of GEOS-Chem version 11-01[19], we quantify the contributions of individual emission source regions on near-surface $PM_{2.5}$ mass concentrations worldwide in each trade scenario. Given the expensive computational costs of GEOS-Chem, we aggregate the 31 GTAP regions into 13 emission source regions (see the mapping in Supplementary Table 4). Largely following previous studies[3,6], the 13 regions are designed based on their economic status and geographical proximity.

$PM_{2.5}$ species simulated by the model include SIOA (including sulfate, nitrate, and ammonium), BC, primary organic aerosol (POA), SOA, anthropogenic dust, natural dust, and sea salt. SIOA, BC, POA, and SOA are derived from both anthropogenic and natural processes. Anthropogenic dust represents dusty particles emitted from industrial and transportation activities (i.e., chimneys of factories and pipes of vehicles). Natural dust and sea salt are emitted from natural processes.

In this study, we only analyze the changes in trade-related (and scenario-dependent) anthropogenic SIOA, BC, and POA concentrations from one trade scenario to another. Emissions from residential activities and private transport remain unchanged across the trade scenarios, so do their impacts on ambient pollutant concentrations. Due to lack of data, anthropogenic dust is also kept constant across the trade scenarios. We do not include the trade scenario-related change in SOA concentrations, which are simulated poorly by current-generation models[69]. Natural SIOA, BC, POA, dust, and sea salt remain unchanged across the trade scenarios.

The all-emission simulation of GEOS-Chem accounts for the impacts of all anthropogenic and natural emissions on $PM_{2.5}$ worldwide in 2014. The simulation is run from June 2013 through December 2014, with the first seven months in 2013 used for model spin-up.

GEOS-Chem is driven by the year-specific GEOS-FP assimilated meteorology from the NASA Global Modeling and Assimilation Office (GMAO). The model is run with the full $O_x$-$NO_x$-VOC-CO-$HO_x$ gaseous chemistry and online aerosol calculations on a 2.5° longitude × 2° latitude grid with 47 vertical layers, and each of the 10 lowest layers are about 130 m thick. Model convection follows the relaxed Arakawa–Schubert scheme[70]. Vertical mixing in the planetary boundary layer employs a non-local scheme implemented by Lin et al.[71]. Dry deposition follows Wesely[72], with a number of modifications[73], for gases and Zhang et al.[74] for aerosols. Wet scavenging of soluble gases and aerosols follows Liu et al.[75], with updates for BC.

Online calculation of SIOA employs the ISOROPIA-II thermodynamic equilibrium model[76], with updates by Zhang et al.[77] on catalytic heterogeneous sulfate formation and Heald et al.[78] on nitrate formation. Uptake of the hydroperoxyl radical on aerosols follows Lin et al. and Ni et al.[79–81]. Anthropogenic aromatics are represented by an increase in propene emissions[79–81]. The mass of POA is assumed to be 1.8 times that of primary organic carbon to account for oxygen atoms contained[3]. Calculation of SOA is parameterized by Pye and Seinfeld[82].

The all-emission simulation uses the CEDS+Xia inventory for global anthropogenic emissions of $NO_x$, $SO_2$, $NH_3$, NMVOC, CO, BC, and POA. Emissions of anthropogenic dust are taken from the MEIC inventory over China, and are assumed to be zero in other countries. Aircraft emissions are taken from AEIC[83] for 2005. International shipping emissions are taken from ICOADS[84] for CO and $NO_x$, from ARCTAS[85,86] for $SO_2$ globally, and from EMEP[87] for $SO_2$ over European waters. Biomass burning emissions follow the GFED4 inventory[88]. Soil $NO_x$ emissions follow Hudman et al.[89]. For lightning $NO_x$ emissions, flash rates are

calculated based on the cloud-top height and constrained by climatological satellite observations[90], and the vertical profile of emitted $NO_x$ follows Ott et al.[91]. Biogenic emissions of NMVOC follow the MEGAN v2.1 model[92]. Natural dust particles are emitted with the DEAD scheme[93–95]. The parameterization of sea salt emissions follows Jaégle et al.[96].

Based on the CEDS+Xia inventory, we further conduct multiple sensitivity simulations based on the zero-out method[2,3,6,65], to estimate the impacts of each region's anthropogenic pollutant emissions on $PM_{2.5}$ concentrations worldwide. We conduct 13 sensitivity simulations (one for each source region), in which anthropogenic emissions of $NO_x$, $SO_2$, $NH_3$, CO, NMVOC, BC, and POA in each region are removed. All other model setups are the same as in the all-emission simulation. The difference between the all-emission simulation and each sensitivity simulation represents the contribution of that region to $PM_{2.5}$ worldwide. In addition, we conduct another sensitivity simulation, in which global anthropogenic emissions of air pollutants are excluded, to represent the natural contribution to the total $PM_{2.5}$. Because we have no robust method to allocate emissions form international shipping and aviation into specific regions, emissions from these sectors are kept unchanged in all sensitivity simulations. Similar to the all-emission simulation, these sensitivity simulations are run from June 2013 through December 2014, with the first seven months in 2013 used for model spin-up.

GEOS-Chem simulations of $PM_{2.5}$ have been validated by Zhang et al.[3], Wang et al.[97], and many other studies, by comparisons with ground, satellite and airborne measurements worldwide. Here we briefly compare the all-emission simulation to the satellite-derived surface $PM_{2.5}$ data from Van Donkelaar et al.[20]. The satellite-derived $PM_{2.5}$ data are estimated by combining satellite retrieved aerosol optical depth and GEOS-Chem, with further calibration based on global ground-based $PM_{2.5}$ observations and geographically weighted regression. The satellite-derived data are re-gridded from its original resolution (0.1° longitude × 0.1° latitude) to match the model resolution. Modeled $PM_{2.5}$ concentration is the sum of SIOA, BC, POA, SOA, dust (2 × DST1 + 0.38 × DST2), and sea salt (SALA). DST1, DST2, and SALA are the names of respective aerosol species in the model contributing to $PM_{2.5}$, and only 38% of DST2 particle mass belong to $PM_{2.5}$. Considering the large underestimate of natural dust by GEOS-Chem[97,98], the simulated concentrations of fine natural dust particles (DST1) are scaled by a factor of 2 prior to the comparison.

Supplementary Fig. 10 compares the simulated, population-weight $PM_{2.5}$ concentrations with the satellite-derived data for individual regions. Each data point represents a model grid cell. For each grid cell of a region (e.g., China), population weighting is done by multiplying the $PM_{2.5}$ concentration of that grid cell by its fractional contribution to the averaged (over the grid cells) population of that region. Supplementary Fig. 10 shows that the simulated results are consistent with the observations, with $R^2$ of 0.82–0.99 and relative mean biases of 2.5–13.0% across the regions.

We use the sum of anthropogenic $PM_{2.5}$ contributed by each region and by global natural emissions (Eq. (6)) as the reference "total $PM_{2.5}$", which is used later as a basis to evaluate the changes in $PM_{2.5}$ and associated premature mortality from one trade scenario to another. This method, instead of using the $PM_{2.5}$ concentrations in the all-emission simulation as the reference, removes the slight effect of chemical nonlinearity in source attribution[3,65].

$$C_{p,i}^b = C_{n,p,i}^b + \sum_{r=1}^{13} C_{r,p,i}^b. \tag{6}$$

$$C_i^b = \sum_p C_{p,i}^b. \tag{7}$$

Here, the superscript b denotes the base case for 2014. The subscript p denotes the $PM_{2.5}$ species; r denotes the anthropogenic source region; n denotes the natural contribution; and i denotes the location (i.e., a model grid cell). $C_{r,p,i}^b$ represents the derived near-surface mass concentration of each $PM_{2.5}$ species at each location contributed by anthropogenic emissions (from all sectors) in region r. $C_{r,p,i}^b$ is derived by subtracting the all-emission simulation by each sensitivity simulation with anthropogenic emissions in the respective source region excluded. $C_{n,p,i}^b$, which is produced from the sensitivity simulation with global anthropogenic emissions excluded, represents the natural $PM_{2.5}$ concentration.

We also use model simulation results to establish the chemical efficiency ($CE_{r,p,i}^b$) of the atmosphere in converting emissions in each region to ambient $PM_{2.5}$ concentrations worldwide

$$E_{r,p'}^b = \sum_s E_{s,r,p'}^b. \tag{8}$$

$$CE_{r,p,i}^b = \frac{C_{r,p,i}^b}{E_{r,p'}^b}. \tag{9}$$

Here, the subscript p' denotes the emitted species ($NO_x + SO_2 + NH_3$, BC, or POA), and p denotes the respective $PM_{2.5}$ species (SIOA, BC, or POA). The subscript i denotes the location (i.e., a model grid cell). The superscript b denotes the base case for 2014. $E_{r,p'}^b$ represents the total anthropogenic emission of species p' in region r. $CE_{r,p,i}^b$ represents the chemical efficiency specific to each source region and $PM_{2.5}$ species, and it remains the same from one trade scenario to another. Following Wang et al.[97], for SIOA, the chemical efficiency is calculated by dividing the concentration of SIOA by the sum of emissions of $NO_x$ (expressed in terms of

nitrate), $SO_2$ (expressed in terms of sulfate) and $NH_3$ (expressed in terms of ammonium), considering the thermodynamic equilibrium of these species. See Supplementary Fig. 4 for more details.

For each scenario, the total $PM_{2.5}$ is contributed by four components: (1) natural aerosols, (2) anthropogenic dust and anthropogenic SOA, (3) anthropogenic but trade scenario-independent SIOA, BC, and POA (i.e., from residential and private vehicle emissions), and (4) trade scenario-dependent SIOA, BC, and POA. Only the last component varies from one trade to another.

To calculate the trade scenario-dependent SIOA, BC, and POA for each trade scenario and source region, we use the prescribed chemical efficiency $CE_{r,p,i}^b$ to convert the scenario- and source region-specific anthropogenic pollutant emissions to respective gridded concentrations worldwide ($C_{r,p,i}^{c,t}$ and $C_{r,i}^{c,t}$ in Eqs. (10) and (11)).

$$C_{r,p,i}^{c,t} = CE_{r,p,i}^b \times E_{r,p'}^{c,t}. \tag{10}$$

$$C_{r,i}^{c,t} = \sum_p C_{r,p,i}^{c,t}. \tag{11}$$

Here, $C_{r,p,i}^{c,t}$ only accounts for trade scenario-dependent anthropogenic SIOA, BC, and POA that vary across the individual trade scenarios. The subscript r denotes the source region, p the $PM_{2.5}$ species (SIOA, BC, or POA), and i the grid cell. The superscript c denotes the trade scenario, and t indicates that this concentration is trade scenario-dependent and is accounted for here.

For the other three $PM_{2.5}$ components that do not vary with trade scenarios, their sum is calculated as follows

$$C_i^{c,o} = C_i^b - \sum_r C_{r,i}^{b,t}. \tag{12}$$

Here, the superscript b represents the base case in 2014 (i.e., Scenario ATR), and o indicates the sum of the other three components. Thus, for the total $PM_{2.5}$ in each trade scenario

$$C_i^c = \sum_r C_{r,i}^{c,t} + C_i^{c,o}. \tag{13}$$

Prior to calculating the health impacts of $PM_{2.5}$, we eliminate the systematic bias in modeled $PM_{2.5}$ concentrations related to errors in model physics and chemistry and errors in emission inputs. Simultaneously, we reproject the $PM_{2.5}$ concentrations from a 2.5° longitude × 2° latitude grid to a 0.1° longitude × 0.1° latitude grid. We first calculate the ratio of the satellite-derived $PM_{2.5}$ concentrations to the modeled $PM_{2.5}$ in the all-emission simulation (Eq. (14)), and then apply the ratio to all trade scenarios (Eqs. (15) and (16)). This procedure ensures that the scenario-specific results are corrected to allow a more accurate health impact estimate.

$$R_j^b = C_j^m / C_i^b. \tag{14}$$

$$C_j'^c = R_j^b \times C_i^c. \tag{15}$$

$$C_{r,j}'^{c,t} = R_j^b \times C_{r,i}^{c,t}. \tag{16}$$

Here, $C_j^m$ represents the satellite-based $PM_{2.5}$ concentration at a 0.1° × 0.1° grid cell j. $C_j'^c$ represents the adjusted total $PM_{2.5}$ concentration at a 0.1° × 0.1° grid cell j, with respect to the pre-adjusted total $PM_{2.5}$ ($C_i^c$) at a 2.5° × 2° grid cell i, in each trade scenario. The center of the finer grid cell j is within the coarser grid cell i. $C_{r,j}'^{c,t}$ represents the adjusted, trade scenario-dependent $PM_{2.5}$ concentration (summed over SIOA, BC, and POA) at each 0.1° × 0.1° grid cell contributed by each source region in each trade scenario.

**Premature deaths due to ambient $PM_{2.5}$ exposure.** We use the GEMM developed by Burnett et al.[21] to estimate $PM_{2.5}$-induced premature deaths in each trade scenario. The GEMM model represents an update upon the IER model used in GBD studies[99]. Both GEMM and IER account for five causes of mortality: ischemic heart disease, stroke, chronic obstructive pulmonary disease, lung cancer, and lower respiratory infections (LRIs). The accounting method in GEMM based on five individual causes is referred to as GEMM 5COD. The GEMM also offers an alternative accounting method (GEMM NCD+LRI) that combines all non-communicable diseases and LRIs[21].

The main text presents our estimated $PM_{2.5}$ induced mortality results based on the GEMM NCD+LRI method. Results based on GEMM 5COD and IER are also presented in Supplementary Data 2 for comparison.

We first apply the above pollution-health models to the adjusted total $PM_{2.5}$ concentrations in each scenario ($C_j'^c$) to derive $PM_{2.5}$-related premature deaths worldwide on a 0.1° longitude × 0.1° latitude grid ($D_j^c$). Detailed models and parameters to calculate $D_j^c$ are presented in Supplementary Data 2. The country-based baseline mortality data for each disease are from the GBD 2016 health data. The gridded population data on a 0.1° × 0.1° spatial resolution are also taken from GBD 2016 health data. To estimate the age-specific health impacts, we employ the country-based age structure from the Unite Nations population data, with the assumption that the age-structure remains unchanged within each region. Based on the country-based baseline mortality, the population data and the age-structure data, we calculate, grid cell by grid cell, the age-specific baseline mortality rate

which equals to the baseline mortality divided by the total population in specific age. When applying the gridded baseline mortality rate data to our health impacts calculation, we assume that the baseline mortality rates remain unchanged across the trade scenarios.

We then use the widely-used direct proportion approach[3,100–102] to assign the fraction of mortality caused by trade scenario-dependent $PM_{2.5}$ (summed over SIOA, BC, and POA) contributed by each source region in each trade scenario ($D_{r,j}^{c,t}$ in Eq. (17)). The direct proportion approach assumes that the contribution of one source to the disease burden of air pollution is directly proportional to its share of the total $PM_{2.5}$ concentration[3,100–102].

$$D_{r,j}^{c,t} = D_j^c \times \frac{C_{r,j}'^{c,t}}{C_j'^c}. \quad (17)$$

Supplementary Data 2 compares our global mortality results in Scenario ATR (which represent the actual situation in 2014) with those by Burnett et al.[21]. There is a slight difference (20%) in global mortality. This is in part because we use an updated version of baseline mortality data upon Burnett et al. In addition, we calculate the mortality for individual grid cells based on their $PM_{2.5}$ concentrations, instead of applying the national average $PM_{2.5}$ concentration to the pollution-health response model, as done by Burnett et al.[21].

**Uncertainty estimates**. Our study is subject to uncertainties from a few sources. First, the GTAP model calculates the changes in global and regional economies from one equilibrium state to another, without considering the temporal (dynamic) evolution of the economies. This means that the model results cannot be compared directly to the economic changes shown in the real economic statistical data. Nonetheless, our model results are consistent with independent economic estimates for various stages of the Sino-US trade war (Supplementary Table 1), which provides confidence in using GTAP for trade scenario analyses.

Second, estimates of emissions are subject to errors in the amount of activity data (e.g., the amount of coal burnt) and emission factors (e.g., the amount of emission per unit of coal burnt)[103]. The overall uncertainty in $CO_2$ emissions is relatively small (within 5% for industrialized countries and within 5–15% for developing regions)[62,104], compared to the uncertainty in air pollutant emissions. We assign the same errors to $CO_2$ emissions in all scenarios.

Third, estimates of air pollutant emissions are affected by errors in emission factors, which rely on the estimate of the level of end-of-pipe emission control, and errors in activity data. The uncertainties in CEDS[17] and Xia et al.[18,42–44] inventories are discussed in detail elsewhere. We adopt the error estimates from previous work for the 13 regions studied here[3,6]. These errors approximately range from 10 to 170% depending on the pollutant and region (Supplementary Fig. 1). For health impact calculations, these errors are implicit in the derivation of the σ2 error below.

Fourth, implementation of the different levels of trade restrictions may affect the energy efficiency and energy source (e.g., coal and solar) in each region and sector. This means that the emission intensity for a given sector may change from one trade scenario to another. This information is partly lost due to our sectoral aggregation. For example, we only have one sector for "Electricity" and thus the fuel mix change cannot be accounted for. A higher level of disaggregation would have the fuel mix changes endogenously included. Although one could exogenously include some sort of efficiency improvement based on extrapolation of previous trends, the approach is subject to the availability of historical data and the appropriateness of extrapolation. Thus we assume that for each region and species, emission intensity of a given sector does not change across the trade scenarios. Nonetheless, the overall emission intensity (i.e., total emission divided by total output from all sectors) is allowed to change because of the change in sectoral output structure (Supplementary Fig. 7). This simplified approach may lead to an additional uncertainty in the calculated emissions. For each scenario other than ATR, the uncertainty is tentatively assigned as σ1 = 5% (one standard deviation), given the amount of fractional change in the global GDP from one scenario to another. σ1 = 0 for Scenario ATR which uses the actual economic data in 2014.

Fifth, as discussed in previous studies[3,6], GEOS-Chem simulations are subject to errors in emissions and model representations of atmospheric chemical and physical processes such as dry deposition, wet scavenging, transport, and formation of secondary aerosols. A full evaluation of model uncertainties is computationally prohibitive[3,6]. However, GEOS-Chem simulations of $PM_{2.5}$ have been validated by comparisons with a comprehensive set of observations[3], and have been adjusted in this study to match the satellite-based $PM_{2.5}$ data. Thus, following Zhang et al.[3], we use the normalized root-mean-square deviation (NRMSD) between the modeled (in the all-emission simulation) and the satellite-based population-weighted $PM_{2.5}$ concentrations to represent the overall model errors for each region (See Supplementary Fig. 10). The error is referred to as σ2 (one standard deviation), which accounts for the combined effects of random errors in emissions and errors in model representations of atmospheric processes.

Sixth, for each trade scenario, we use prescribed region- and species-specific chemical efficiency data to convert from pollutant emissions to ambient concentrations. The chemical efficiency data are calculated based on model sensitivity simulations, and are assumed to be unchanged across the individual trade scenarios. This assumption may lead to slight errors for SIOA due to the thermodynamic interdependence between sulfate, nitrate and ammonium. Nonetheless, the magnitudes of chemical efficiency calculated by GEOS-Chem are

comparable to results from other models[105,106]. An additional uncertainty related to the use of chemical efficiency arises from the fact that within each of the 13 emission source regions in GEOS-Chem simulations, there may be multiple GTAP regions, due to the mapping from 31 GTAP regions to 13 GEOS-Chem regions. This mean that the spatial pattern of emissions within each of the 13 regions may slightly change from one trade scenario. For each scenario other than ATR, we tentatively assign a σ3 = 15% error (one standard deviation) due to use of chemical efficiency. σ3 = 0 for Scenario ATR, whose model results are the same as the base case of GEOS-Chem driven by the emissions in 2014.

Seventh, the pollution-health models used here (GEMM NCD+LRI, GEMM 5COD, and IER) are subject to large errors in linking pollution exposure, specific diseases, and premature death. In particular, the two GEMM models do not consider the potential differences in toxicity between the individual $PM_{2.5}$ components. The accuracy of pollution-health models is also limited by the amount of cohort studies used to build the models[21]. To build the IER model, cohort studies related to not just ambient pollution but also indoor pollution and smoking are used[101]. To evaluate the uncertainty from pollution-health models, we calculate the mortality based on each of GEMM NCD+LRI, GEMM 5COD, and IER. Furthermore, for each model we calculate the 95% CI for the estimated mortality data, through a bootstrap method which incorporates both sampling and model shape uncertainty. The corresponding error (one standard deviation) is referred to as σ4 (one standard deviation).

The overall uncertainty in the mortality data for each trade scenario is estimated as the sum in quadrature of σ1–σ4. Error results are expressed as 95% CI in the main text. Although σ2 and σ4 are dominant sources of error, they are derived from causes that do not depend on trade scenarios. Thus, σ2 and σ4 are not relevant when discussing the relative change in premature mortality from one trade scenario to another.

## Data availability
All data used here are cited in the text. The datasets generated during this study are available from the corresponding authors.

## Code availability
All computer codes generated during this study are available from the corresponding authors upon reasonable request.

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

## Acknowledgements

This study is supported by the National Natural Science Foundation of China (41775115). K.F. and K.H. acknowledge the funding support from University of Maryland's BSOS Dean's Research Initiative. K.H. acknowledges support from the Czech Science Foundation under the project VEENEX (GA ČR no. 16-17978S). Y.L. acknowledges support from the the National Natural Science Foundation of China (71974186) and National Key Research and Development Program of China (Grant No. 2016YFA0602500).

## Author contributions

J.L. conceived the research. J.L., K.F. and M.D. designed the research. M.D., L.C. and Y.L. performed the research. J.L. and M.D. designed the scenarios. K.F. and Y.L. provided socioeconomic data. Y.Z. provided Chinese emissions data. R.M. and A.v.D. provided satellite-based PM$_{2.5}$ data. L.C. and M.D. calculated emissions. L.C. and R.N. performed atmospheric simulations. L.C. and J.W. calculated chemical efficiencies. L.C., H.K. and M.D. performed simulations of health impacts. J.L., M.D., L.C. and K.F. led the analysis with inputs from K.H., R.M., H.W., M.L. and Q.L. J.L., M.D. and L.C. led the writing with inputs from Y.L. All authors discussed the results and commented on the manuscript.

## Competing interests
The authors declare no competing interests.
