## [Peer Review File · Nature Communications]

Reviewers' comments:

Reviewer #1 (Remarks to the Author):

This paper presents an exceedingly simplistic view of how international trade affects the environment. This is problematic because it appears that the paper's reliance on unrealistic assumptions as to how international trade affects pollution means that the analysis is incorrect; it greatly overstates the health effects of international trade.

The key issue here is the paper's reliance of fixed emission intensities to translate the effects of trade into changes in pollution emissions. This is an extremely serious flaw with the analysis; the assumption flies in the face of over 25 years of research in economics that highlights the fact that emission intensities will respond endogenously to trade. These endogenous responses mean that pollution levels can actually fall with trade, and indeed, the existing empirical evidence in economics suggest that this is, in fact, the case (see, for example, the work of Antweiler et. al. (Am. Econ. Rev., 2001)). This means pollution levels would likely be lower with free trade, not higher, meaning the paper is greatly overstating the potential health effects of trade. For the paper's results to be considered credible, the analysis needs to take the possibility of endogenous changes in emission intensity seriously and incorporate the various mechanisms that the economics literature has proposed to explain these endogenous responses (such as income induced policy responses and offshoring, for example). The authors should consult the literature reviews by Copeland and Taylor (J. Econ. Lit., 2004) and Cherniwchan et. al. (Annu. Rev. Econ., 2017)) for overviews of the various mechanisms that have been proposed.

Reviewer #2 (Remarks to the Author):

The paper uses an integrated modelling framework to assess the effects of trade restrictions on economic growth, air pollution and mortality. Below, I provide some comments that can improve the quality of the manuscript.

Comments:

- It is not clear to the reader if the sources for estimating air pollutant emission factors across sectors and regions are compatible among them. The authors should mention the different sources used and the compatibility among them.
- It also not clear how the air pollution factors/intensities are estimated in the study. Since these are not derived from the GTAP model the study misses the indirect and induced impact of consumption on air pollution. These factors change only with the change of the economic output from the trade scenarios, i.e., there are simple direct air pollutant emission factors.
- It is not clear which is the base year of the input-output tables used in this version of GTAP to better understand how well the structure of the economy can capture the effects of trade scenarios.
- The paper very nicely indicates the impact of trade restrictions in economy, air pollution and mortality terms at territorial level. However, there is no discussion on the relative contribution of the economic sectors on air pollution and mortality and mainly which sectors (industry, transportation, agriculture) are affected/affect most economic growth, air pollution and mortality. There are several studies discussing those issues, e.g., doi: 10.1016/j.scitotenv.2019.01.410; doi.org/10.5194/acp-16-1895-2016
- In LN 251-262 the authors contrast GFT with an additional scenario GFTT to capture the effect of reducing emission intensities in developing countries. However, this reduction either through stricter environmental legislation or technical progress will affect the competitiveness of the developing economies which will subsequently change the terms of trade between countries; a loop that is not considered in the study.
- The study concludes that the high emission intensities in developing countries are the biggest issue to be solved. But are developing countries the largest polluters also in absolute terms? More

specific policy recommendations linking the impact of trade liberalisation/restrictions on economic growth and air pollution with sustainable growth paths at sectoral and territorial level could further increase the quality of the paper.

Reviewer #3 (Remarks to the Author):

Lin et al. integrated different interdisciplinary models to calculate carbon and pollution health impacts of trade restrictions. The work will be of interest to the broader community and the wider scientific field, and is novel to influence thinking in the field. This is a fairly comprehensive work as evident from the Main Text, Supplementary Discussions and Extended Data.

Following are points that need to be addressed:

Major comments:

1. One of the key uncertainties of the paper is that trade-scenario related changes in secondary organic aerosols (SOA) are not included, although secondary inorganic aerosols are included. The authors suggest in Supplemental Text Line 287 that this is because SOA is simulated poorly by current generation models.

I find that this reasoning is not convincing, especially because SOA is an important part of PM_{2.5} and is needed to produce model-measurement agreement, although the authors adjust GEOS-Chem PM₂₅ with satellite derived estimates. One approach might be to vary the trade-dependent anthropogenic SOA by scaling unmeasured SOA precursors (semi-volatile and intermediate volatility organic vapors) as a function of POA or NMVOC e.g. see references 1, 2, 3. Although SOA is uncertain, this scaling would allow a mechanistic understanding of how variation in trade scenarios affect anthropogenic SOA similar to POA, since SOA precursors are scaled with respect to POA or NMVOCs.

2. Another source of uncertainty is introduced by the author's use of chemical efficiency to calculate how different trade scenarios affect PM₂₅ concentrations. It would be much better to actually run different GEOS-Chem simulations for each trade scenario-associated emissions explicitly, rather than using chemical efficiency from a base GEOS-Chem simulation. I understand that chemical efficiency was needed since GEOS-Chem simulations are computationally expensive. But the authors need to at least demonstrate how this affects their results/conclusions by conducting additional GEOS-Chem simulations for a different trade scenario explicitly.

Minor comments:

1. Supplemental Figure 4: Recommend using different colors for symbols to distinguish different regions on the plot. For e.g. to aid understanding of discussions on lines 111-112 in the main text, it would be good to know which symbols are for China, West Europe and USA.

2. Lines 198-201: This sentence needs to talk about developing regions rather than developed regions since 79-83% of global mortality reduction is attributed to developing regions (only 17-22%) to developed regions.

3. Lines 206-209: Why is relative reduction in economic output from GFT to GTB smaller in more emission-intensive sectors? I would think that trade wars would reduce emission intensive sectors like road transportation, chemical industries, electricity production more drastically and so their economic output will be more strongly affected. Please explain.

4. Line 259: The figure for GFTT is actually Extended Data Figure 8. It is incorrectly referred at Extended Data Figure 7.

5. Line 259-262: The text talks about % reduction. But that is not directly shown in the Extended

Data figure 8. Instead, it has to be inferred from the hatched bars in the figure. For example, in sub Saharan Africa and South Asia, it's particularly hard to see. The % reduction could be plotted as a 3rd panel plot separately.

6. The authors have described their methodology and sources of uncertainty in the supplementary text. But the uncertainty discussion is very important and needs to also be included in the Methods section of the main text. In addition, a plot of relative uncertainties for PM25 mortality should be included. This is needed to motivate future work in this field.

References:

1. Shrivastava M, et al. Urban pollution greatly enhances formation of natural aerosols over the Amazon rainforest. *Nature Communications* 10, 1046 (2019).
2. Shrivastava M, et al. Global transformation and fate of SOA: Implications of low-volatility SOA and gas-phase fragmentation reactions. *J Geophys Res-Atmos* 120, 4169-4195 (2015).
3. Shrivastava M, Lane TE, Donahue NM, Pandis SN, Robinson AL. Effects of gas particle partitioning and aging of primary emissions on urban and regional organic aerosol concentrations. *J Geophys Res-Atmos* 113, (2008).

Reviewer #1:

This paper presents an exceedingly simplistic view of how international trade affects the environment. This is problematic because it appears that the paper's reliance on unrealistic assumptions as to how international trade affects pollution means that the analysis is incorrect; it greatly overstates the health effects of international trade.

The key issue here is the paper's reliance of fixed emission intensities to translate the effects of trade into changes in pollution emissions. This is an extremely serious flaw with the analysis; the assumption flies in the face of over 25 years of research in economics that highlights the fact that emission intensities will respond endogenously to trade. These endogenous responses mean that pollution levels can actually fall with trade, and indeed, the existing empirical evidence in economics suggest that this is, in fact, the case (see, for example, the work of Antweiler et. al. (Am. Econ. Rev., 2001)). This means pollution levels would likely be lower with free trade, not higher, meaning the paper is greatly overstating the potential health effects of trade. For the paper's results to be considered credible, the analysis needs to take the possibility of endogenous changes in emission intensity seriously and incorporate the various mechanisms that the economics literature has proposed to explain these endogenous responses (such as income induced policy responses and offshoring, for example).

The authors should consult the literature reviews by Copeland and Taylor (J. Econ. Lit., 2004) and Cherniwchan et. al. (Annu. Rev. Econ., 2017)) for overviews of the various mechanisms that have been proposed.

Re: The reviewer interpreted that we used fixed emission intensity and argued that had emission intensity been allowed to change with trade scenarios, the health impacts results would be reversed. In particular, the reviewer stated that “pollution levels can actually fall with trade”. Here we clarify that the overall emission intensity (as a weighted average from all sectors) is changed from one scenario to another, even though the emission intensity for each sector is not. We also do not agree that “pollution levels can actually fall with trade” is a persistent fact on the global and regional scales studied here, although it may be true for certain specific firms. Instead, our paper specifically points out the possibility that, on the global and regional scales, trade development could adversely affect public health under certain conditions (i.e., by not reducing emission intensity in each sector), and also offers a plausible alternative scenario to avoid such an unnecessary and undesirable outcome.

1. For each pollutant and region, the overall emission intensity (as a weighted average from all sectors) is changed from one scenario to another in this study. The overall emission intensity depends on two factors: 1) the industry structure and 2) the emission intensity of each industry sector. In our study, the change in industry structure is accounted for by using the GTAP CGE to calculate the output changes in individual sectors due to trade restrictions. The emission intensity in each sector is fixed for reasons explained in the third point below. Our results suggest that the overall emission intensity of a region and pollutant is lower in a freer trade scenario and higher in a more trade-restricted case. See the main text (line 205-207): *“This result also indicates enhanced emission intensities of CO₂ and pollutants in an anti-trade world represented by Scenario GTB, compared to GFT.”*

2. The reviewer argued that “pollution levels can actually fall with trade”, and offered three references to support the argument. Among the three references, two are fairly old (Antweiler et. al. 2001; Copeland and Taylor, 2004)^{1,2} and do not reflect the current trade and environmental situations, and the third reference (Cherniwchan et. al., 2017)³ is for firm level studies, the scale of which does not match the regional and global scales being studied here. In contrast, several recent studies⁴⁻⁷ have shown substantial transboundary pollution embedded in trade – in particular, the current trade activities have allowed polluted industries to move from developed to developing regions, which tend to increase the emission intensities of these industries and the pollution levels of exporting regions.

Many arguments, including those in the references the reviewer provided, that trade may lead to less pollution are based on the Environmental Kuznets Curve (EKC) hypothesis. Recent regional and global scale studies, however, do not support the existence of a general EKC phenomenon for air pollutant emissions. For example, the regional scale study by Ru et al (2018)⁸ showed that 1) although regions' SO₂ emissions from the power and industrial sectors largely follow an EKC pattern, there is substantial noise in the EKC curve, 2) SO₂ emissions from other sectors do not show an EKC, and 3) black carbon emissions do not exhibit any EKC pattern in any sector.

3. We assumed that the emission intensity of each industry sector is fixed, for several reasons :

- a) For the regional and global scale analysis in this study, there are no reliable, quantitative data on how the emission intensity of each sector and region would respond to trade-related economic changes. As we wrote in Supplementary Section S2, to account for the change in sectoral emission intensity, "*Although one could exogenously include some sort of efficiency improvement based on extrapolation of previous trends, the approach is subject to the availability of historical data and the appropriateness of extrapolation.*" The limited data from firm-level studies (suggested by the reviewer) on endogenous emission intensity are less useful given the global scope of our study.
- b) Thus, instead of assuming certain hypothetical relationship between emission intensity of each sector and the level of trade restriction (and economic prosperity) on the regional and global scales, we have elected to fix the emission intensity of each sector for all scenarios except GFTT. Based on these non-GFTT scenarios, we show that the pollution level and health impact would be higher in a freer trade scenario and lower in a more trade-restricted world, if the emission intensity of each sector is not changed. We then design an alternative GFTT scenario which, on top of the free-trade scenario (GFT), includes reductions in sectoral emission intensities (by globally collective, collaborative actions to cut regional pollution, especially in developing regions). In contrast to GFT, this GFTT scenario offers a win-win pathway to allow trade to be beneficial for both economy and environment, which is in line with the

reviewer's argument that trade may lead to an improved environment.

We have added a new paragraph in the revised main text (the second last paragraph, lines 275-291) on the uncertainties and limitations of our study, including the use of fixed emission intensity for each sector.

Reviewer #2:

The paper uses an integrated modelling framework to assess the effects of trade restrictions on economic growth, air pollution and mortality. Below, I provide some comments that can improve the quality of the manuscript.

Re: Thank you for your helpful comments and suggestions, which have been incorporated in our revised manuscript.

Comments:

- It is not clear to the reader if the sources for estimating air pollutant emission factors across sectors and regions are compatible among them. The authors should mention the different sources used and the compatibility among them.

Re : In the revised manuscript, we have added a description of data sources and a reference to SI for details. See the main text (lines 65-67): *“a customized emission inventory derived from the Community Emission Database System (CEDS) and Xia et al. (see Supplementary Section S1.5 for details)”*.

The two inventories have been carefully matched to eliminate the compatibility issue. For detailed data sources and compatibility, we showed in Supplementary Section S1.5 that:

“The GTAP model does not provide emissions of air pollutants. Thus, we calculate anthropogenic air pollutant emissions for each trade scenario based on prescribed sector-, region- and pollutant-specific emission intensity data (that remain unchanged across the trade scenarios) and scenario-specific economic output from GTAP.

We derive the sector-, region- and species-specific emission intensities by combining 1) a customized emission inventory (CEDS+Xia, see below) in 2014 and 2) economic output data in the GTAP database in 2014”.

We use the monthly gridded (0.5° longitude \times 0.5° latitude) Community Emission Database System (CEDS) inventory for global anthropogenic emissions of gaseous (SO_2 , NO_x , NH_3 , NMVOC , and CO) and primary aerosol (BC and OC) pollutants worldwide in 2014, with 54 sectors in 152 regions. The CEDS inventory has a globally consistent and reproducible methodology applied to all pollutants and includes updated emission factors. It provides very detailed sectoral emission information, which is essential for this study. The inventory is being used by the Coupled Model Inter-comparison Project Phase 6 (CMIP6, a main model support for the Inter-governmental Panel on Climate Change Sixth Assessment Report) and many other studies.

The CEDS inventory uses regional emission data over the US (1990–2014), Canada (1990–2013), Europe (1980–2013), China (2008, 2010 and 2012), and other regions to revise its initial global methodology. Over the past decade, the amount of emissions in China varied significantly from one year to another due to implementation of stringent emission control measures as well as the changing economy and fossil fuel consumption. Thus, we replace the seasonal and spatial patterns of Chinese SO_2 , NO_x , CO , BC and POA emissions in CEDS by those in the MEIC inventory in 2014. We further scale Chinese annual SO_2 , NO_x , CO , BC and POA emissions in CEDS to match those developed by Xia et al. (2016), which account for the pollution control measures more comprehensively and may better represent the actual emissions in China in 2014. Since the CEDS and Xia et al. inventories contain different sectors, we conduct a sector mapping procedure (Supplementary Table 4). Hereafter, we refer to this hybrid inventory as CEDS+Xia.

The CEDS+Xia inventory contains 54 sectors in 152 countries/regions. Of these sectors, 8 belong to energy production, 23 belong to industry, 8 belong to transportation, 4 belong to residential, 5 belong to agriculture, 4 belong to waste treatment, and the remaining 2 represent other un-specified processes that are associated with very few emissions. The inventory is a global monthly gridded dataset at a 0.5° longitude \times 0.5° latitude resolution, beneficial for subsequent GEOS-Chem simulations (Sect. 1.6).”

We also discussed emission errors in Supplementary Section S2:

“Second, estimates of emissions are subject to errors in the amount of activity data (e.g., the amount of coal burnt) and emission factors (e.g., the amount of emission per unit of coal burnt). The overall uncertainty in CO₂ emissions is relatively small (within 5% for industrialized countries and within 5%–15% for developing regions), compared to the uncertainty in air pollutant emissions. We assign the same errors to CO₂ emissions in all scenarios.

Third, estimates of air pollutant emissions are affected by errors in emission factors, which rely on the estimate of the level of end-of-pipe emission control, and errors in activity data. The uncertainties in CEDS and Xia et al. inventories are discussed in detail elsewhere. We adopt the error estimates from previous work for the 13 regions studied here. These errors approximately range from 10% to 170% depending on the pollutant and region (Extended Data Figure 2). For health impact calculations, these errors are implicit in the derivation of the σ_2 error below.”

- It also not clear how the air pollution factors/intensities are estimated in the study.

Since these are not derived from the GTAP model the study misses the indirect and induced impact of consumption on air pollution. These factors change only with the change of the economic output from the trade scenarios, i.e., there are simple direct air pollutant emission factors.

Re: In this study, emission intensity is calculated by the sectoral emission (CEDS+Xia, production-based) divided by sectoral output for each sector, pollutant and region. These emission intensity values are then applied to each trade scenario. Please refer to our reply to reviewer 1 for detailed reasoning of our calculation method.

The calculation of emission intensity is described in detail in Supplementary Section S1.5.3:

“Based on the CEDS+Xia inventory, we derive a prescribed dataset for emission intensity that varies across the sectors, regions and pollutant species, by dividing the CEDS+Xia emissions by the economic output data in the GTAP database for 2014. This emission intensity dataset is used and remains unchanged in all trade scenarios.

We convert the CEDS+Xia emissions for 152 regions and 54 sectors to 141 regions and 57 sectors according to the original GTAP setup, and then to 31

regions and 20 sectors to match those in our trade scenario analyses. The mapping details are shown in Supplementary Table 7a and b.

Subsequently, we calculate emission intensity for each pollutant in each of the 20 sectors and 31 regions:

$$F_{s,r,p}^b = E_{s,r,p}^b / X_{s,r}^b \quad (3)$$

Here the subscripts s , r and p denote the sector, region and pollutant species, respectively. The superscript b denotes the base year (2014) that has CEDS+Xia emissions ($E_{s,r,p}^b$) and economic output in the GTAP database for 2014 ($X_{s,r}^b$). ”

We have clarified in the main text (lines 70-73) that “For each sector, pollutant and region, emission intensity is calculated by the sectoral emission from the inventory divided by the actual sectoral output in 2014. The emission intensity value is then applied to the sectoral output of each trade scenario to obtain the respective emission.”

Consumption based pollution analysis is out of the scope of this study, which is focused on production based pollution changes due to trade restrictions.

- It is not clear which is the base year of the input-output tables used in this version of GTAP to better understand how well the structure of the economy can capture the effects of trade scenarios.

Re: We use the latest GTAP model with the base year of 2014, in line with the year of this study. We have added this information in the revised main text (lines 63-65): “we take an interdisciplinary approach to integrating the latest standard Global Trade Analysis Project (GTAP, version 10 data base for 2014) Computable General Equilibrium model14-16 for global trade and economy”

- The paper very nicely indicates the impact of trade restrictions in economy, air pollution and mortality terms at territorial level. However, there is no discussion on the relative contribution of the economic sectors on air pollution and mortality and mainly which sectors (industry, transportation, agriculture) are affected/affect most economic growth, air pollution and mortality. There are several studies discussing those issues, e.g., doi: 10.1016/j.scitotenv.2019.01.410; doi.org/10.5194/acp-16-1895-2016

Re: In this study, we mainly focus on the carbon and pollution impacts of changes in the total output (i.e., summed over all sectors) due to trade restrictions. Thus we did not focus on a sector-based analysis.

Per your suggestion, we have added more discussion in the main text (lines 203-214) on how sectoral output changes affect the overall emission intensity: *“This result also indicates enhanced (sectorally averaged) emission intensities of CO2 and pollutants in an anti-trade world represented by Scenario GTB, compared to GFT. This is because individual economic sectors have different emission intensities^{9,10} and different responses to economic shocks from trade restrictions. Sectors with high emission intensities such as Electricity and Road Transport are often not directly affected by trade restrictions, since they do not produce goods for trade. By comparison, sectors with low emission intensities such as Wearing Apparel and Textiles are often directly affected by trade restrictions. As shown in Extended Data Figure 7, the relative reduction in economic output from GFT to GTB is smaller in more emission-intensive sectors, resulting in increased relative contributions of emission-intensive sectors to global output.”*

- In LN 251-262 the authors contrast GFT with an additional scenario GFTT to capture the effect of reducing emission intensities in developing countries. However, this reduction either through stricter environmental legislation or technical progress will affect the competitiveness of the developing economies which will subsequently change the terms of trade between countries; a loop that is not considered in the study.

Re: We agree and have added a discussion on this issue in the revised main text (lines 263-267) : *“Enhancing the environmental regulations in developing regions means an economic burden (at least in the beginning) to the industries that may affect their competitiveness and subsequently alter the inter-regional trade, which factor is not fully accounted for in GFTT. Nonetheless, enhancing the financial and technological support may reduce the initial shock to the developing economies.”*

- The study concludes that the high emission intensities in developing countries are the biggest issue to be solved. But are developing countries the largest polluters also in absolute terms? More specific policy recommendations linking the impact of trade liberalisation/restrictions on economic growth and air pollution with sustainable growth paths at sectoral and territorial level could further increase the quality of the paper.

Re : Developing countries are indeed the largest emitters. We stated in the main text (line 237-239) that *“Changes in CO₂, pollutant emissions and mortality are dominated by those in developing regions with higher emission intensities (Extended Data Figure 6, Supplementary Figure 4).”*

Nonetheless, we find that the dominant cause of the high emissions in developing regions are their high emission intensities. In terms of economic output, developing regions are still behind developed regions. More importantly, the increasing global emissions with trade liberalization is primarily due to growth of developing economies in part because of the movement of polluted industries from developed to developing regions^{4,7}. Thus, we think cutting emission intensities of developing regions is the key to mitigating the global pollution.

The aim of this study is to analyze the potential (likely unforeseen) environmental consequences of trade restrictions/liberalization, to better inform trade and environmental policymakers. Although this study is not a heavily policy oriented analysis, we discuss the potential of achieving an economic and environmental win-win goal by both improving trade liberalization and enhancing global environmental collaborations (in Scenario GFTT). We have added in the end of the revised manuscript that (lines 301-303) :*“To this end, our study offers insight for policymakers who might consider to better address in trade negotiations the potential environmental implications of trade to ensure sustainable growth on regional and global scales.”*

Reviewer #3:

Lin et al. integrated different interdisciplinary models to calculate carbon and pollution health impacts of trade restrictions. The work will be of interest to the broader community and the wider scientific field, and is novel to influence thinking in the field. This is a fairly comprehensive work as evident from the Main Text, Supplementary Discussions and Extended Data.

Re: Thank you very much for your positive comments on our manuscript.

Following are points that need to be addressed:

Major comments:

1. One of the key uncertainties of the paper is that trade-scenario related changes in secondary organic aerosols (SOA) are not included, although secondary inorganic aerosols are included. The authors suggest in Supplemental Text Line 287 that this is because SOA is simulated poorly by current generation models.

I find that this reasoning is not convincing, especially because SOA is an important part of PM_{2.5} and is needed to produce model-measurement agreement, although the authors adjust GEOS-Chem PM₂₅ with satellite derived estimates. One approach might would be to vary the trade-dependent anthropogenic SOA by scaling unmeasured SOA precursors (semi-volatile and intermediate volatility organic vapors) as a function of POA or NMVOC e.g. see references 1, 2, 3

Although SOA is uncertain, this scaling would allow a mechanistic understanding of how variation in trade scenarios affect anthropogenic SOA similar to POA, since SOA precursors are scaled with respect to POA or NMVOCs.

Re: This study analyzes anthropogenic SIOA (including nitrate, sulfate and ammonium), black carbon, and primary organic aerosols, which together contribute the dominant portion of anthropogenic PM_{2.5} pollution relevant to trade restrictions. Thus, we believe that exclusion of SOA does not affect the general conclusion of this study.

We agree that a mechanistic understanding of how variation in trade scenario affects anthropogenic SOA is important, and that omitting anthropogenic SOA is a limitation of our study. SOA is produced from both anthropogenic and biogenic (natural) VOC emissions. In anthropogenic emissions-heavy regions like China, (anthropogenic + biogenic) SOA contribute 20% of the total PM_{2.5} mass concentration¹¹⁻¹³, and anthropogenic sources are more important than biogenic sources. In regions like the US, SOA is mainly of biogenic origin, although anthropogenic NO_x and SO₂ may have indirect influences on the formation of biogenic SOA¹⁴⁻¹⁶. This study is focused on anthropogenic pollution, and thus anthropogenic SOA may be more relevant for China and other developing regions.

We have elected not to include anthropogenic SOA, following previous studies^{5,17,18}, in part because of the difficulty to properly allocate anthropogenic VOC emissions to detailed sectors. In particular, large fractions of VOC emissions are from fugitive processes (i.e., evaporation from solvents and fuels), for which sectoral allocation in emission inventories remains a major challenge and is subject to a large uncertainty^{4-6,19}. For other pollutants (NO_x, SO₂, BC, POA) considered in this study,

anthropogenic emissions are from fuel combustion and can be allocated to different sectors with a much higher accuracy.

Another factor precluding the inclusion of SOA in our study is the relatively poor performance of the version of SOA scheme in GEOS-Chem. Globally, the model produces a too small fraction of SOA (about 3%) in the total OA (i.e., POA + SOA), far lower than observations (about 60%) (Schroder et al., 2018)²⁰. The performance is also poor over China, i.e., $R < 0.5$ and $NMB = -40\% - -80\%$ based on comparisons with in situ observations (Miao & Chen et al., IGC9 poster)²¹. The chemical reaction formula suggested by the reviewer is used in WRF-Chem and is not straightforward to be adopted in our simulation here. For example, the formula for semi-volatile SOA formation due to oxidation of unmeasured SOA precursors from Jathar et al. (2014)²², which is used in the paper¹⁶ suggested by the reviewer, is not suitable for this version of GEOS-Chem due to mismatch in SOA scheme, species and other issues.

We have added a new paragraph in the revised main text (the second last paragraph, line 275-291) on the uncertainties and limitations of our study, including the exclusion of SOA. We have also cited the references suggested by the reviewer.

2. Another source of uncertainty is introduced by the author's use of chemical efficiency to calculate how different trade scenarios affect PM_{2.5} concentrations. It would be much better to actually run different GEOS-Chem simulations for each trade scenario-associated emissions explicitly, rather than using chemical efficiency from a base GEOS-Chem simulation. I understand that chemical efficiency was needed since GEOS-Chem simulations are computationally expensive. But the authors need to at least demonstrate how this affects their results/conclusions by conducting additional GEOS-Chem simulations for a different trade scenario explicitly.

Re: We chose to use chemical efficiencies instead to running all simulations based on two considerations: 1) our previous study suggested a very linear relationship between emissions and PM_{2.5} mass concentrations^{6,23}, and 2) we wanted to minimize computational costs.

Per your suggestion, we test this linearity issue by running GEOS-Chem to simulate the SIOA, BC and POA concentrations due to China's anthropogenic emissions in Scenario GTB (Global Trade Barrier), by using the calculated emissions to drive GEOS-Chem simulation; note that China is one of the 13 regions studied here. We then compare the results of this sensitivity simulation with those by using chemical efficiencies. For BC and POA, results from running GEOS-Chem and from using the

chemical efficiencies are virtually identical, which is expected considering that BC and POA are chemically inert in GEOS-Chem. For SIOA, the difference between directly running GEOS-Chem and using the chemical efficiencies is only about 8% both over China (Figure 1) and globally. These results support our choice of using chemical efficiencies.

Figure 1: Trade scenario-dependent SIOA mass concentrations (a) derived using chemical efficiency, (b) simulated by GEOS-Chem, and (c) their difference. Unit is $\mu\text{g}/\text{m}^3$.

In our uncertainty discussion in SI Section S2, we have included the effect of using chemical efficiencies:

“Sixth, for each trade scenario, we use prescribed region- and species-specific chemical efficiency data to convert from pollutant emissions to ambient concentrations. The chemical efficiency data are calculated based on model sensitivity simulations, and are assumed to be unchanged across the individual trade scenarios. This assumption may lead to slight errors for SIOA (about 8%) due to the thermodynamic interdependence between sulfate, nitrate and ammonium. Nonetheless, the magnitudes of chemical efficiency calculated by GEOS-Chem are comparable to results from other models. An additional uncertainty related to the use of chemical efficiency arises from the fact that within each of the 13 emission source regions in GEOS-Chem simulations, there may be multiple GTAP regions, due to the mapping from 31 GTAP regions to 13 GEOS-Chem regions. This mean that the spatial pattern of emissions within each of the 13 regions may slightly change from one trade scenario. For each scenario other than ATR, we tentatively assign a $\sigma_3 = 15\%$ error (one standard deviation) due to use of chemical efficiency. $\sigma_3 = 0$ for Scenario ATR, whose model results

are the same as the base case of GEOS-Chem driven by the emissions in 2014.”

We have added a new paragraph in the revised main text (the second last paragraph, lines 275-291) on the uncertainties and limitations of our study, including the use of chemical efficiencies.

Minor comments:

1. Supplemental Figure 4: Recommend using different colors for symbols to distinguish different regions on the plot. For e.g. to aid understanding of discussions on lines 111-112 in the main text, it would be good to know which symbols are for China, West Europe and USA.

Re: We have added the regional names in Supplemental Figure 4.

2. Lines 198-201: This sentence needs to talk about developing regions rather than developed regions since 79-83% of global mortality reduction is attributed to developing regions (only 17-22%) to developed regions.

Re: We have revised this sentence by describing developing regions, see lines 198-203 in the main text:

“Overall, about 52%–64% of global CO2 emission reduction and 78%–83% of global mortality reduction from GFT to ATR, TW1, TW2 and GTB occur in developing regions (China, rest of East Asia, Economies in Transition, Latin America and Caribbean, Middle East and North Africa, South Asia, South-East Asia and Pacific, and Sub-Saharan Africa), with the rest in developed regions (Extended Data Figure 6).”

3. Lines 206-209: Why is relative reduction in economic output from GFT to GTB smaller in more emission-intensive sectors? I would think that trade wars would reduce emission intensive sectors like road transportation, chemical industries, electricity production more drastically and so their economic output will be more strongly affected. Please explain.

Re: Industries with high emission intensity such as Electricity and Road Transport are usually not directly affected by trade restrictions, since they do not produce goods for trade. Electricity and Road Transport are affected indirectly by trade restrictions because of changes in other industrial (e.g., Wearing Apparel and Textiles) production that requires electricity and transportation. In contrast, industries with low emission intensity such as Wearing Apparel and Textiles are directly affected by trade restrictions.

4. Line 259: The figure for GFTT is actually Extended Data Figure 8. It is incorrectly referred at Extended Data Figure 7.

Re: We have corrected it in revised main text (lines 271-272): *“The respective regional reductions are substantial (Extended Data Figure 8).”*

5. Line 259-262: The text talks about % reduction. But that is not directly shown in the Extended Data figure 8. Instead, it has to be inferred from the hatched bars in the figure. For example, in sub Saharan Africa and South Asia, it's particularly hard to see. The % reduction could be plotted as a 3rd panel plot separately.

Re: We have added the percentage values in the Extended Data Figure 8.

6. The authors have described their methodology and sources of uncertainty in the supplementary text. But the uncertainty discussion is very important and needs to also be included in the Methods section of the main text. In addition, a plot of relative uncertainties for PM25 mortality should be included. This is needed to motivate future work in this field.

Re: We have added a new paragraph (as the second last paragraph of the main text, lines 275-291) to summarize the uncertainty discussion shown in detail in Supplementary Section S2:

“Several sources of uncertainty and limitation exist in our study, as detailed in Supplementary Section S2. The standard GTAP model is an equilibrium model that does not simulate the temporal evolution of the economies. Emission data are subject to large errors especially for air pollutants. We do not account for the change in emission intensity of a given sector from one trade scenario to another, although the overall emission intensity (i.e., total emission divided by total output from all sectors) is allowed to change because of the change in sectoral output structure. GEOS-Chem simulations are subject to errors in emissions and model representations of atmospheric chemical and physical processes. In particular, secondary organic aerosols^{16,24,25} are not simulated here. Using chemical efficiencies to calculate pollution levels for each trade scenario further introduces a minor source of error. For each scenario, a major source of error arises from the pollution-health exposure models used here. Quantitative estimates of these errors are given in Supplementary Section S2. The overall error results are expressed as 95% CI in the main text. Although errors in emissions and pollution-health models are dominant, they are derived from causes that do not depend on trade scenarios, and are thus not relevant when

discussing the relative change in premature mortality from one trade scenario to another.”

Extended Data Figure 3 shows the uncertainty in mortality estimate. Also, Supplementary Table 9 includes more detailed results for uncertainty estimates and comparisons with previous studies.

References:

- 1 Antweiler, W., Copeland, B. R. & Taylor, M. S. Is Free Trade Good for the Environment? *Nber Working Papers* **91**, 877-908 (2001).
- 2 Copeland, B. R. & Taylor, M. S. Trade, Growth, and the Environment. *Journal of Economic Literature* **42**, 7-71 (2004).
- 3 Cherniwchan, J., Copeland, B. R. & Taylor, M. S. Trade and the Environment: New Methods, Measurements, and Results. *Annual Review of Economics* **9**, 59-85, doi:10.1146/annurev-economics-063016-103756 (2017).
- 4 Lin, J. *et al.* China's international trade and air pollution in the United States. *Proceedings of the National Academy of Sciences* **111**, 1736-1741, doi:10.1073/pnas.1312860111 (2014).
- 5 Lin, J. *et al.* Global climate forcing of aerosols embodied in international trade. *Nature Geoscience* **9**, 790 (2016).
- 6 Zhang, Q. *et al.* Transboundary health impacts of transported global air pollution and international trade. *Nature* **543**, 705 (2017).
- 7 Jakob, M. & Marschinski, R. Interpreting trade-related CO₂ emission transfers. *Nature Climate Change* **3**, 19, doi:10.1038/nclimate1630 (2012).
- 8 Ru, M., Shindell, D. T., Seltzer, K. M., Tao, S. & Zhong, Q. The long-term relationship between emissions and economic growth for SO₂, CO₂, and BC. *Environmental Research Letters* **13**, 124021 (2018).
- 9 Giannakis, E. *et al.* Exploring the economy-wide effects of agriculture on air quality and health: Evidence from Europe. *Science of The Total Environment* **663**, 889-900, doi:https://doi.org/10.1016/j.scitotenv.2019.01.410 (2019).
- 10 Aksoyoglu, S., Prévôt, A. S. H. & Baltensperger, U. Contribution of Ship Emissions to the Concentration and Deposition of Pollutants in Europe: Seasonal and Spatial Variation. *Atmospheric Chemistry & Physics* **16**, 1895-1906 (2016).
- 11 Fu, T. M. *et al.* Carbonaceous aerosols in China: top-down constraints on primary sources and estimation of secondary contribution. *Atmospheric Chemistry and Physics* **12**, 2725-2746, doi:10.5194/acp-12-2725-2012 (2012).
- 12 Chen, Q., Fu, T.-M., Hu, J., Ying, Q. & Zhang, L. Modelling secondary organic aerosols in China. *National Science Review* **4**, 806-809, doi:10.1093/nsr/nwx143 (2017).

- 13 Xie, Y. Z. *et al.* Characteristics of chemical composition and seasonal variations of PM_{2.5} in Shijiazhuang, China: Impact of primary emissions and secondary formation. *Science of the Total Environment* **677**, 215-229, doi:10.1016/j.scitotenv.2019.04.300 (2019).
- 14 Zhang, H. *et al.* Monoterpenes are the largest source of summertime organic aerosol in the southeastern United States. *Proceedings of the National Academy of Sciences* **115**, 2038-2043, doi:10.1073/pnas.1717513115 (2018).
- 15 Shilling, J. E. *et al.* Enhanced SOA formation from mixed anthropogenic and biogenic emissions during the CARES campaign. *Atmospheric Chemistry and Physics* **13**, 2091-2113, doi:10.5194/acp-13-2091-2013 (2013).
- 16 Shrivastava, M. *et al.* Urban pollution greatly enhances formation of natural aerosols over the Amazon rainforest. *Nature Communications* **10**, doi:10.1038/s41467-019-08909-4 (2019).
- 17 Paulot, F. & Jacob, D. J. Hidden Cost of U.S. Agricultural Exports: Particulate Matter from Ammonia Emissions. *Environmental Science & Technology* **48**, 903-908, doi:10.1021/es4034793 (2014).
- 18 Guan, D. *et al.* The socioeconomic drivers of China's primary PM_{2.5} emissions. *Environmental Research Letters* **9**, 024010, doi:10.1088/1748-9326/9/2/024010 (2014).
- 19 Hoesly, R. M. *et al.* Historical (1750–2014) anthropogenic emissions of reactive gases and aerosols from the Community Emissions Data System (CEDS). *Geoscientific Model Development* **11**, 369-408 (2018).
- 20 Schroder, J. C. *et al.* Sources and Secondary Production of Organic Aerosols in the Northeastern United States during WINTER. *Journal of Geophysical Research: Atmospheres*, doi:10.1029/2018jd028475 (2018).
- 21 Miao, R. Q. & Chen, Q. Simulation of organic aerosol in China. *Poster presentation at the 9th International GEOS-Chem Meeting, Cambridge* (2019).
- 22 Jathar, S. H. *et al.* Unspeciated organic emissions from combustion sources and their influence on the secondary organic aerosol budget in the United States. *Proceedings of the National Academy of Sciences* **111**, 10473-10478, doi:10.1073/pnas.1323740111 (2014).
- 23 Wang, J. *et al.* Socioeconomic and atmospheric factors affecting aerosol radiative forcing: Production-based versus consumption-based perspective. *Atmospheric Environment* **200**, 197-207, doi:https://doi.org/10.1016/j.atmosenv.2018.12.012 (2019).
- 24 Shrivastava, M. K., Donahue, N. M., Pandis, S. N. & Robinson, A. L. Effects of gas particle partitioning and aging of primary emissions on urban and regional organic aerosol concentrations. *Journal of Geophysical Research Atmospheres* **113**, - (2008).
- 25 Shrivastava, M. *et al.* Global transformation and fate of SOA: Implications of Low Volatility SOA and Gas-Phase Fragmentation Reactions: Global modeling of SOA. *Journal of Geophysical Research Atmospheres* **120**, 4169-4195 (2015).

REVIEWERS' COMMENTS:

Reviewer #2 (Remarks to the Author):

The authors have addressed my comments; very nice integrated work

Reviewer #3 (Remarks to the Author):

The authors have addressed most of my major comments.

I agree with the authors that anthropogenic VOC emissions and also SVOC/IVOC emissions (unmeasured) are a major source of uncertainty. Also, if anthropogenic SOA constitutes just 20% of PM_{2.5} in urban areas in China, excluding it may not produce major errors with respect to their analysis.

But I disagree with the author's justification of not including anthropogenic SOA in GEOS-Chem based on mismatch with species and chemical schemes. SOA from unmeasured SVOCs/IVOCs from anthropogenic pollution can easily be included as additional species/chemical reactions. This has been done for another global modeling study:

<https://agupubs.onlinelibrary.wiley.com/doi/full/10.1002/2014JD022563>

But if anthropogenic SOA is not as important for China, this is not critical to their paper. However, I recommend the authors cite additional observational studies over urban China to substantiate their claim that this SOA component is just 20% of PM_{2.5}.

In future studies, anthropogenic SOA can and should be included for completeness.

Reviewer #2:

The authors have addressed my comments; very nice integrated work.

Re: Thanks for your positive response.

Reviewer #3:

The authors have addressed most of my major comments. I agree with the authors that anthropogenic VOC emissions and also SVOC/IVOC emissions (unmeasured) are a major source of uncertainty. Also, if anthropogenic SOA constitutes just 20% of PM_{2.5} in urban areas in China, excluding it may not produce major errors with respect to their analysis. But I disagree with the author's justification of not including anthropogenic SOA in GEOS-Chem based on mismatch with species and chemical schemes. SOA from unmeasured SVOCs/IVOCs from anthropogenic pollution can easily be included as additional species/chemical reactions. This has been done for another global modeling study:

<https://agupubs.onlinelibrary.wiley.com/doi/full/10.1002/2014JD022563>

But if anthropogenic SOA is not as important for China, this is not critical to their paper. However, I recommend the authors cite additional observational studies over urban China to substantiate their claim that this SOA component is just 20% of PM_{2.5}. In future studies, anthropogenic SOA can and should be included for completeness.

Re: In China, SOA (anthropogenic plus biogenic) constitutes about 15-30% of the surface mass of PM_{2.5} during haze events¹⁻³. Among total SOA, the anthropogenic SOA constitutes only less than 50%⁴. In Other regions like USA and Europe, the contribution of anthropogenic SOA is only approximately 10%⁵⁻⁷. Therefore, exclusion of anthropogenic SOA did not produce major errors on the robustness of this study. We have added some references to substantiate our claim about anthropogenic SOA in the main text. See the main text (lines 239-240): “*In particular, secondary organic aerosols³³⁻³⁵ are not simulated here, considering the relative small contribution of anthropogenic secondary organic aerosols to the total PM_{2.5}³⁶⁻³⁹.*”

We also agree with the reviewer’s opinion that the anthropogenic SOA is worth discussing. The effects of anthropogenic SOA will be paid high attention in our future studies.

References

- 1 Chen, Q., Fu, T.-M., Hu, J., Ying, Q. & Zhang, L. Modelling secondary organic aerosols in China. *National Science Review* **4**, 806-809, doi:10.1093/nsr/nwx143 (2017).
- 2 Fu, T. M. *et al.* Carbonaceous aerosols in China: top-down constraints on primary sources and estimation of secondary contribution. *Atmospheric Chemistry and Physics* **12**, 2725-2746, doi:10.5194/acp-12-2725-2012 (2012).
- 3 Li, Y. J. *et al.* Real-time chemical characterization of atmospheric particulate matter in China: A review. *Atmospheric Environment* **158**, 270-304, doi:<https://doi.org/10.1016/j.atmosenv.2017.02.027> (2017).
- 4 Jiang, F. *et al.* Regional modeling of secondary organic aerosol over China using WRF/Chem. *Journal of Aerosol Science* **43**, 57-73, doi:<https://doi.org/10.1016/j.jaerosci.2011.09.003> (2012).
- 5 Kleindienst, T. E. *et al.* Estimates of the contributions of biogenic and anthropogenic hydrocarbons to secondary organic aerosol at a southeastern US location. *Atmospheric Environment* **41**, 8288-8300, doi:<https://doi.org/10.1016/j.atmosenv.2007.06.045> (2007).
- 6 Gelencsér, A. *et al.* Source apportionment of PM_{2.5} organic aerosol over Europe: Primary/secondary, natural/anthropogenic, and fossil/biogenic origin. *J. Geophys. Res.-Atmos.* **112**, doi:10.1029/2006jd008094 (2007).
- 7 Volkamer, R. *et al.* Secondary organic aerosol formation from anthropogenic air pollution: Rapid and higher than expected. *Geophysical Research Letters* **33**, doi:10.1029/2006gl026899 (2006).